# REINFORCEMENT LEARNING WITH RANDOM DELAYS

**Yann Bouteiller**[*]
Polytechnique Montreal
yann.bouteiller@polymtl.ca

**Simon Ramstedt**[*]
Mila, McGill University
simonramstedt@gmail.com

**Giovanni Beltrame**
Polytechnique Montreal

**Christopher Pal**
Mila, Polytechnique Montreal

**Jonathan Binas**
Mila, University of Montreal

## ABSTRACT

Action and observation delays commonly occur in many Reinforcement Learning applications, such as remote control scenarios. We study the anatomy of randomly delayed environments, and show that partially resampling trajectory fragments in hindsight allows for off-policy multi-step value estimation. We apply this principle to derive Delay-Correcting Actor-Critic (DCAC), an algorithm based on Soft Actor-Critic with significantly better performance in environments with delays. This is shown theoretically and also demonstrated practically on a delay-augmented version of the MuJoCo continuous control benchmark.

## 1  INTRODUCTION

This article is concerned with the Reinforcement Learning (RL) scenario depicted in Figure 1, which is commonly encountered in real-world applications (Mahmood et al., 2018; Fuchs et al., 2020; Hwangbo et al., 2017). Oftentimes, actions generated by the agent are not immediately applied in the environment, and observations do not immediately reach the agent. Such environments have mainly been studied under the unrealistic assumption of constant delays (Nilsson et al., 1998; Ge et al., 2013; Mahmood et al., 2018). Here, prior work has proposed different planning algorithms which naively try to undelay the environment by simulating future observations (Walsh et al., 2008; Schuitema et al., 2010; Firoiu et al., 2018).

We propose an off-policy, planning-free approach that enables low-bias and low-variance multi-step value estimation in environments with random delays. First, we study the anatomy of such environments in order to exploit their structure, defining Random-Delay Markov Decision Processes (RDMDP). Then, we show how to transform trajectory fragments collected under one policy into trajectory fragments distributed according to another policy. We demonstrate this principle by deriving a novel off-policy algorithm (DCAC) based on Soft Actor-Critic (SAC), and exhibiting greatly improved performance in delayed environments. Along with this work we release our code, including a wrapper that conveniently augments any OpenAI gym environment with custom delays.

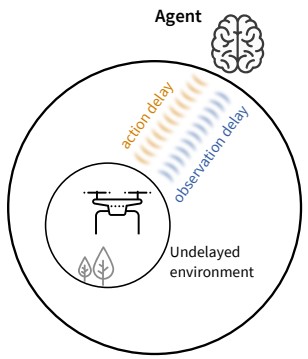

Figure 1: A delayed environment can be decomposed into an undelayed environment and delayed communication dynamics.

## 2  DELAYED ENVIRONMENTS

We frame the general setting of real-world Reinforcement Learning in terms of an *agent*, random *observation delays*, random *action delays*, and an *undelayed environment*. At the beginning of each time-step, the agent starts computing a new action from the most recent available delayed observation. Meanwhile, a new observation is sent and the most recent delayed action is applied in the undelayed environment. Real-valued delays are rounded up to the next integer time-step.

---

[*]equal contribution

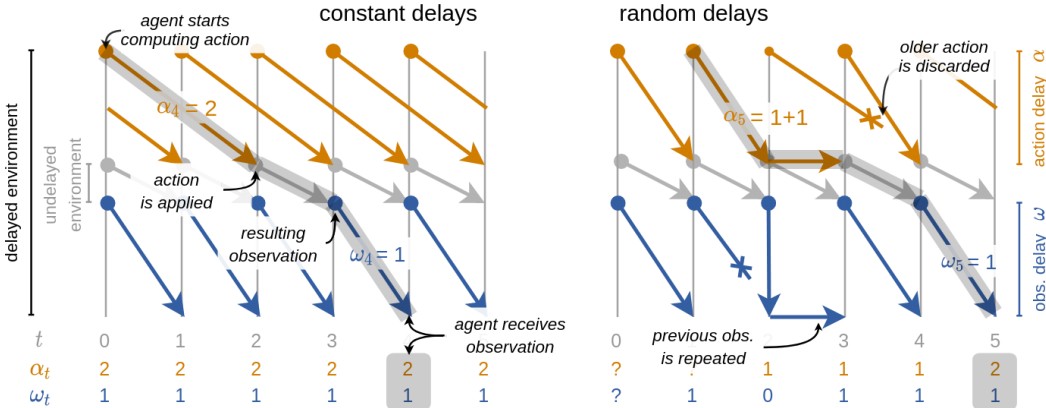

Figure 3: Influence of actions on delayed observations in delayed environments.

For a given delayed observation $s_t$, the *observation delay* $\omega_t$ refers to the number of elapsed time-steps from when $s_t$ finishes being captured to when it starts being used to compute a new action. The *action delay* $\alpha_t$ refers to the number of elapsed time-steps from when the last action influencing $s_t$ starts being computed to one time-step before $s_t$ finishes being captured. We further refer to $\omega_t + \alpha_t$ as the *total delay* of $s_t$.

As a motivating illustration of real-world delayed setting, we have collected a dataset of communication delays between a decision-making computer and a flying robot over WiFi, summarized in Figure 2. In the presence of such delays, the naive approach is to simply use the last received observation. In this case, any delay longer than one time-step violates the Markov assumption, since the last sent action becomes an unobserved part of the current state of the environment. To overcome this issue, we define a Markov Decision Process that takes into account the communication dynamics.

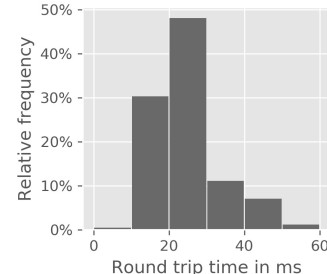

Figure 2: Histogram of real-world WiFi delays.

## 2.1 RANDOM DELAY MARKOV DECISION PROCESSES

To ensure the Markov property in delayed settings, it is necessary to augment the delayed observation with at least the last $K$ sent actions. $K$ is the combined maximum possible observation and action delay. This is required as the oldest actions along with the delayed observation describe the current state of the undelayed environment, whereas the most recent actions are yet to be applied (see Appendix C). Using this augmentation suffices to ensure that the Markov property is met in certain delayed environments. On the other hand, it is possible to do much better when the delays themselves are also part of the state-space. First, this allows us to model self-correlated delays, e.g. discarding outdated actions and observations (see Appendix A.1). Second, this provides useful information to the model about how old an observation is and what actions have been applied next. Third, knowledge over the total delay allows for efficient credit assignment and off-policy partial trajectory resampling, as we show in this work.

**Definition 1.** A Random Delay Markov Decision Process $RDMDP(E, p_\omega, p_\alpha) = (X, A, \tilde{\mu}, \tilde{p})$ augments a Markov Decision Process $E = (S, A, \mu, p)$ with:
(1) state-space $X = S \times A^K \times \mathbb{N}^2$, (2) action-space $A$,
(3) initial state distribution $\tilde{\mu}(x_0) = \tilde{\mu}(s, u, \omega, \alpha) = \mu(s)\, \delta(u - c_u)\, \delta(\omega - c_\omega)\, \delta(\alpha - c_\alpha)$,
(4) transition distribution $\tilde{p}(s', u', \omega', \alpha', r' | s, u, \omega, \alpha, a) = f_{\omega - \omega'}(s', \alpha', r' | s, u, \omega, \alpha, a) p_\omega(\omega' | \omega) p_u(u' | u, a)$,

where $s \in S$ is the delayed observation, $u \in A^K$ is a buffer of the last $K$ sent actions, $\omega \in \mathbb{N}$ is the *observation delay*, and $\alpha \in \mathbb{N}$ is the *action delay* as defined above. To avoid conflicting with the subscript notation, we index the action buffers' elements using square brackets. Here, $u_{[1]}$ is the most recent and $u_{[K]}$ is the oldest action in the buffer. We denote slices by $u_{[i:j]} = (u_{[i]}, \ldots, u_{[j]})$ and $u_{[i:-j]} = (u_{[i]}, \ldots, u_{[K-j]})$. We slightly override this notation and additionally define $u_{[0]} = a$.

The constants $c_u \in A^K$ and $c_\omega, c_\alpha \in \mathbb{N}$ initialize $u, \omega, \alpha$, since $\delta$ is the Dirac delta distribution. The transition distribution itself is composed of three parts: (1) The observation delay distribution $p_\omega$ modelling the evolution of observation delays. Note that this density function must represent a discrete distribution (i.e. be a weighted sum of Dirac delta distributions). Furthermore, this process will repeat observations if there are no new ones available. This means that the observation delay can maximally grow by one from one time-step to the next. (2) The transition distribution for the action buffer $p_u(u'|u, a) = \delta(u' - (a, u_{[1:-1]}))$. (3) The distribution $f_\Delta$ describing the evolution of observations, rewards and action delays (Definition 2).

**Definition 2.** For each change in observation delays ($\Delta = \omega - \omega'$) we define a variable step update distribution $f_\Delta$ as

$$f_\Delta(s', \alpha', r' | s, u, \omega, \alpha, a) = \mathbb{E}_{s^*, \alpha^*, r^* \sim f_{\Delta-1}(\cdot | s, u, \omega, \alpha, a)}[p(s', r' - r^* | s^*, u[\overbrace{\omega - \Delta + \alpha'}^{\omega'}]) \, p_\alpha(\alpha'|\alpha^*)]. \quad (1)$$

The base case of the recursion is $f_{-1}(s', \alpha', r' \mid s, u, \omega, \alpha, a) = \delta(s' - s) \, \delta(\alpha' - \alpha) \, \delta(r')$.

Here, $p_\alpha$ is the action delay distribution which, similar to $p_\omega$, must be discrete. The transition distribution of the underlying, undelayed MDP is $p$. The $r' - r*$ term accumulates intermediate rewards in case observations are skipped or repeated (see Appendix A.4). Since the observation delay cannot increment by more than one, $f_{-1}$ is used when $\omega$ is increasing, whereas $f_0$ is used when there is no change in observation delay.

A simple special case of the RDMDP is the constant observation and action delay case with $p_\omega(\omega'|\omega) = \delta(\omega' - c_\omega)$ and $p_\alpha(\alpha'|\alpha) = \delta(\alpha' - c_\alpha)$. Here, the RDMDP reduces to a Constantly Delayed Markov Decision Process, described by Walsh et al. (2008). In this case, the action and observation delays $\alpha, \omega$ can be removed from the state-space as they don't carry information. Examples of RDMDP dynamics are visualized in Figure 3 (see also Appendix C).

## 3 REINFORCEMENT LEARNING IN DELAYED ENVIRONMENTS

Delayed environments as described in Section 2 are specific types of MDP, with an augmented state-space and delayed dynamics. Therefore, using this augmented state-space, traditional algorithms such as Soft Actor-Critic (SAC) (Haarnoja et al., 2018a)(Haarnoja et al., 2018b) will always work in randomly delayed settings. However, their performance will still deteriorate because of the more difficult credit assignment caused by delayed observations and rewards, on top of the exploration and generalization burdens of delayed environments. We now analyze how to compensate for the credit assignment difficulty by leveraging our knowledge about the delays' dynamics.

One solution is to perform *on-policy* multi-step rollouts on sub-trajectories that are longer than the considered delays. On the other hand, on-policy algorithms are known to be sample-inefficient and therefore are not commonly used in real-world applications, where data collection is costly. This motivates the development of *off-policy* algorithms able to reuse old samples, such as SAC.

Intuitively, in delayed environments, one should take advantage of the fact that actions only influence observations and rewards after a number of time-steps relative to the beginning of their computation (the total delay $\omega + \alpha$). Since the delay information is part of the state-space, it can be leveraged to track the action influence through time. However, applying conventional off-policy algorithms in delayed settings leads to the following issue: the trajectories used to perform the aforementioned multi-step backups have been sampled under an outdated policy, and therefore contain outdated action buffers. In this section, we propose a method to tackle this issue by performing *partial trajectory resampling*. We make use of the fact that the delayed dynamics are known to simulate the effect they would have had under the current policy, effectively transforming off-policy sub-trajectories into on-policy sub-trajectories. This enables us to derive a family of efficient off-policy algorithms for randomly delayed settings.

### 3.1 PARTIAL TRAJECTORY RESAMPLING IN DELAYED ENVIRONMENTS

One important observation implied by Figure 3 is that, given the delayed dynamics of RDMDPs, some actions contained in the action buffer of an off-policy state did not influence the subsequent delayed

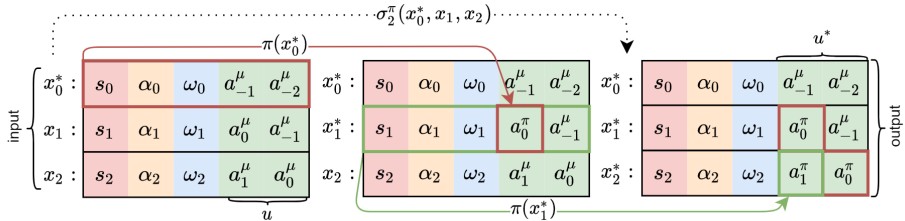

Figure 4: Partial resampling of a small sub-trajectory. The action buffer is recursively resampled according to the current policy $\pi$ (rewards are not modified by $\sigma$ and are omitted here).

observations and rewards for a number of time-steps. Therefore, if an off-policy sub-trajectory is short enough, it is possible to recursively resample its action buffers with no influence on the return. We propose the following transformation of off-policy sub-trajectories:

**Definition 3.** The partial trajectory resampling operator recursively updates action buffers as follows

$$
\sigma_n^\pi \big( \underbrace{s_1^*, u_1^*, \omega_1^*, \alpha_1^*}_{x_1^*}, r_1^*, \tau_{n-1}^* \, | \, x_0^*; \underbrace{s_1, u_1, \omega_1, \alpha_1}_{x_1}, r_1, \tau_{n-1} \big)
$$

$$
= \delta\big( (s_1^*, \omega_1^*, \alpha_1^*, r_1^*) - (s_1, \omega_1, \alpha_1, r_1) \big) \mathbb{E}_{a_0 \sim \pi(\cdot|x_0^*)} \big[ \delta\big(u_1^* - (a_0, u_0^*[1:-1]) \big) \big] \, \sigma_{n-1}^\pi (\tau_{n-1}^* | x_1^*; \tau_{n-1}) \quad (2)
$$

with trivial base case $\sigma_0(x_0^*) = 1$

This operator recursively resamples the most recent actions of each action buffer in an input sub-trajectory $\tau_n$, according to a new policy $\pi$. Everything else stays unchanged. A visual example is provided in Figure 4 with $n = 2$ and an action buffer of two actions. When resampled actions are delayed and would not affect the environment, they do not "invalidate" the sub-trajectory. The resampled trajectories can then be considered on-policy.

**Theorem 1.** The partial trajectory resampling operator $\sigma_n^\pi$ (Def. 3) transforms off-policy trajectories into on-policy trajectories

$$
\mathbb{E}_{\tau_n \sim p_n^\mu(\cdot|x_0)} [\sigma_n^\pi(\tau_n^* | x_0; \tau_n)] = p_n^\pi(\tau_n^* | x_0) \quad (3)
$$

on the condition that none of the delayed observations depend on any of the resampled actions, i.e.

$$
\omega_t^* + \alpha_t^* \geq t \quad (4)
$$

where $t$ indexes the trajectory $\tau_n^* = (s_1^*, u_1^*, \omega_1^*, \alpha_1^*, r_1^*, \ldots, s_n^*, u_n^*, \omega_n^*, \alpha_n^*, r_n^*)$ from 1 to $n$.

The condition in Equation 4 can be understood visually with the help of Figure 3. In the constant delay example it is fulfilled until the third time-step. After that, the observations would have been influenced by the resampled actions (starting with $a_0$).

## 3.2 Multi-step Off-Policy Value Estimation in Delayed Environments

We have shown in Section 3.1 how it is possible to transform off-policy sub-trajectories into on-policy sub-trajectories in the presence of random delays. From this, we can derive a family of efficient off-policy algorithms for the randomly delayed setting. For this matter, we make use of the classic on-policy Monte-Carlo $n$-step value estimator:

**Definition 4.** The n-step state-value estimator is defined as

$$
\hat{v}_n(x_0; \underbrace{x_1^*, r_1^*, \tau_{n-1}^*}_{\tau_n^*}) = r_1^* + \gamma \hat{v}_{n-1}(x_1^*; \, \tau_{n-1}^*) = \sum_{i=1}^{n} \gamma^{i-1} r_i^* + \gamma^n \hat{v}_0(x_n^*). \quad (5)
$$

where $\hat{v}_0$ is a state-value function approximator (e.g. a neural network).

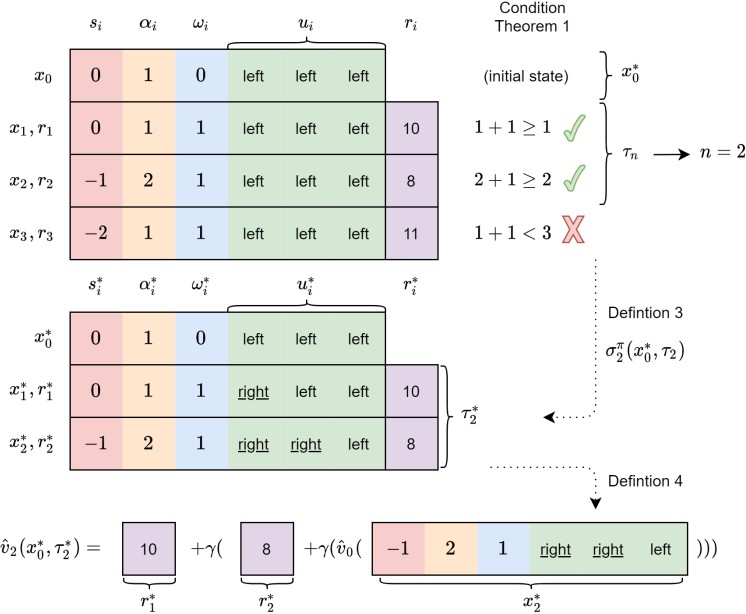

Figure 5: Visual example in a 1D-world with random delays ($K = 3$). The original trajectory has been sampled under the policy $\mu$: 'always go left'. The current policy is $\pi$: 'always go right'.

Indeed, in $\gamma$-discounted RL, performing on-policy n-step rollouts to estimate the value function reduces the bias introduced by the function approximator by a factor of $\gamma^n$:

**Lemma 1.** The n-step value estimator has the following bias:

$$\text{bias}(\hat{v}_n(x_0, \cdot)) = \gamma^n \mathbb{E}_{\ldots, x_n^*, r_n^* \sim p_n^\pi(\cdot|x_0)}[\text{bias}(\hat{v}_0(x_n^*))] \tag{6}$$

A simple corollary of Lemma 1 is that the on-policy n-step value estimator is unbiased when the function approximator $\hat{v}_0$ is unbiased. On the other hand, Theorem 1 provides a recipe for transforming sub-trajectories collected under old policies into actual on-policy sub-trajectories. From a given state in an off-policy trajectory, this is done by applying $\sigma_n^\pi$ to all the subsequent transitions until we meet a total delay ($\omega_i + \alpha_i$) that is shorter than the length of the formed sub-trajectory. Consequently, the transformed sub-trajectory can be fed to the on-policy n-step value estimator, where n is the length of this sub-trajectory. This does not only provide a better value estimate than usual 1-step off-policy estimators according to Lemma 1, but it maximally compensates for the multi-step credit assignment difficulty introduced by random delays. Indeed, the length of the transformed sub-trajectory is then exactly the number of time-steps it took the first action of the sub-trajectory to have an influence on subsequent delayed observations, minus one time-step.

As opposed to other unbiased n-step off-policy methods, such as importance sampling and Retrace (Munos et al., 2016), this method doesn't suffer from variance explosion. This is because the presence of delays allows us to transform off-policy sub-trajectories into on-policy sub-trajectories, so that old samples don't need to be weighted by the policy ratio.

Although we use a multi-step state-value estimator, the same principles can be applied to action-value estimation as well. In fact, the trajectory transformation described in Definition 3 enables efficient off-policy n-step value estimation in any value-based algorithm that would otherwise perform 1-step action-value backups, such as DQN, DDPG or SAC. In the next section, we illustrate this using SAC.

Figure 5 summarizes the whole procedure in a simple 1D-world example. The maximum possible delay is $K = 3$ here, and the agent can only go 'left' or 'right'. An initial augmented state $x_0$ is sampled from the replay memory, along with the 3 subsequent augmented states and rewards. The condition of Theorem 1 is satisfied for $n \leq 2$. It follows that $\tau_n = \tau_2 = (x_1, x_2)$. This off-policy trajectory fragment is partially resampled, which yields the corresponding on-policy trajectory fragment $\tau_n^* = \tau_2^*$. This on-policy trajectory fragment can then be used to compute an unbiased n-step value estimate of the initial state $x_0 = x_0^*$.

## 4 DELAY-CORRECTING ACTOR-CRITIC

We have seen in Section 3 how it is possible, in the delayed setting, to collect *off-policy* trajectories and still use *on-policy* multi-step estimators in an unbiased way, which allows us to compensate for the more difficult credit assignment introduced by the presence of random delays. We now apply this method to derive Delay-Correcting Actor-Critic (DCAC), an improved version of Soft Actor-Critic (Haarnoja et al., 2018a;b) for real-time randomly delayed settings.

### 4.1 VALUE APPROXIMATION

Like SAC, DCAC makes use of the entropy-augmented soft value function (Haarnoja et al., 2018a):

**Lemma 2.** In a RDMDP $(E, p_\omega, p_\alpha)$ the soft value function is:

$$v^{\text{soft}}(x_0^*) = \mathbb{E}_{a \sim \pi(\cdot|x_0^*)}[\mathbb{E}_{x_1^*, r_1^* \sim \tilde{p}(\cdot|x_0^*, a)}[r_1^* + \gamma v^{\text{soft}}(x_1^*)] - \log \pi(a|x_0^*)] \tag{7}$$

It can be estimated by augmenting the reward function in Definition 4 with an entropy reward:

**Definition 5.** The delayed on-policy n-step soft state-value estimator, i.e. the n-step state-value estimator with entropy augmented rewards under the current policy $\pi$, is

$$\hat{v}_n^{\text{soft}}(x_0^*; \tau_n^*) = r_1^* + \gamma \hat{v}_{n-1}^{\text{soft}}(x_1^*; \tau_{n-1}^*) - \mathbb{E}_{a \sim \pi(\cdot|x_0^*)}[\log \pi(a|x_0^*)] \tag{8}$$

where $\hat{v}_0^{\text{soft}}$ is a state-value function approximator (e.g. a neural network).

Given the off-policy trajectory transformation proposed in Section 3, Definition 5 directly gives DCAC's value target. To recap, we sample an initial state $x_0$ ($= x_0^*$) and a subsequent trajectory $\tau_n$ ($= x_1, r_1, \ldots x_n, r_n$) from a replay memory. The sampling procedure ensures that $n$ is the greatest length so that the sampled trajectory $\tau_n$ does not contain any total delay $\omega_i + \beta_i < i$. This trajectory was collected under an *old policy* $\mu$, but we need a trajectory compatible with the *current policy* $\pi$ to use $\hat{v}_n^{\text{soft}}$ in an unbiased way. Therefore, we feed $\tau_n$ to the partial trajectory resampling operator defined in Definition 3. This produces an *equivalent on-policy* sub-trajectory $\tau_n^*$ with respect to the current policy $\pi$ according to Theorem 1, while maximally taking advantage of the bias reduction described by Lemma 1. This partially resampled on-policy sub-trajectory is fed as input to $\hat{v}_n^{\text{soft}}(x_0; \tau_n^*)$, which yields the target used in DCAC's soft state-value loss:

**Definition 6.** The DCAC critic loss is

$$L_v^{\text{DCAC}}(v) = \mathbb{E}_{(x_0, \tau_n) \sim \mathcal{D}} \, \mathbb{E}_{\tau_n^* \sim \sigma_n^\pi(\cdot|x_0; \tau_n)}[(v_\theta(x_0) - \hat{v}_n^{\text{soft}}(x_0; \tau_n^*))^2] \tag{9}$$

where $x_0, \tau_n$ are a start state and following trajectory, sampled from the replay memory, and satisfying the condition of Theorem 1.

### 4.2 POLICY IMPROVEMENT

In addition to using the on-policy n-step value estimator as target for our parametric value estimator, we can also use it for policy improvement. As in SAC we use the reparameterization trick (Kingma & Welling, 2013) to obtain the policy gradient from the value estimator. However, since we use our trajectory transformation and a multi-step value estimator, this involves backpropagation through time in the action buffer.

**Definition 7.** The DCAC actor loss is

$$L_\pi^{\text{DCAC}}(\pi) = -\mathbb{E}_{(x_0, \tau_n) \sim \mathcal{D}} \, \mathbb{E}_{\tau_n^* \sim \sigma_n^\pi(\cdot|x_0; \tau_n)}[\hat{v}_n^{\text{soft}}(x_0; \tau_n^*)] \tag{10}$$

where $x_0, \tau_n$ are a start state and following trajectory, sampled from the replay memory, and satisfying the condition of Theorem 1.

**Proposition 1.** The DCAC actor loss is a less biased version of the SAC actor loss with

$$\text{bias}(L_\pi^{\text{DCAC}}) = \mathbb{E}_n[\gamma^n]\,\text{bias}(L_\pi^{\text{SAC}}) \tag{11}$$

assuming both are using similarly biased parametric value estimators to compute the loss, i.e.

$$\text{bias}(\hat{v}_0^{\text{soft}}(x)) = \mathbb{E}_{a\sim\pi(\cdot|x)}[\text{bias}(\hat{q}_0^{\text{soft}}(x,a))] \tag{12}$$

## 5 EXPERIMENTAL RESULTS

To evaluate our approach and make future work in this direction easy for the RL community, we release as open-source, along with our code, a Gym wrapper that introduces custom multi-step delays in any classical turn-based Gym environment. In particular, this enables us to introduce random delays to the Gym MuJoCo continuous control suite (Brockman et al., 2016; Todorov et al.), which is otherwise turn-based.

**Compared algorithms.** A naive version of SAC would only use the unaugmented delayed observations, which violates the Markov assumption in delayed settings as previously pointed out. Consequently, naive SAC exhibits near-random results in delayed environments. A few such experiments are provided in the Appendix for illustration (Figure 9).

In order to make a fair comparison, all other experiments compare DCAC against SAC in the same RDMDP setting, i.e. all algorithms use the augmented observation space defined in Section 2.1. Since SAC is the algorithm we chose to improve for delayed scenarios, comparing DCAC against it in the same setting provides a like-for-like comparison. We also found it interesting to compare against RTAC (Ramstedt & Pal, 2019). Indeed, DCAC reduces to this algorithm in the special case where observation transmission is instantaneous ($\omega$=0) and action computation and transmission constantly takes one time-step ($\alpha$=1). Whereas DCAC performs variable-length state-value backups with partial trajectory resampling as explained in Section 4 , RTAC performs 1-step state-value backups, and SAC performs the usual 1-step action-value backup described in its second version (Haarnoja et al., 2018b). All hyperparameters and implementation details are provided in Section B of the Appendix.

For each experiment, we perform six runs with different seeds, and shade the 90% confidence intervals.

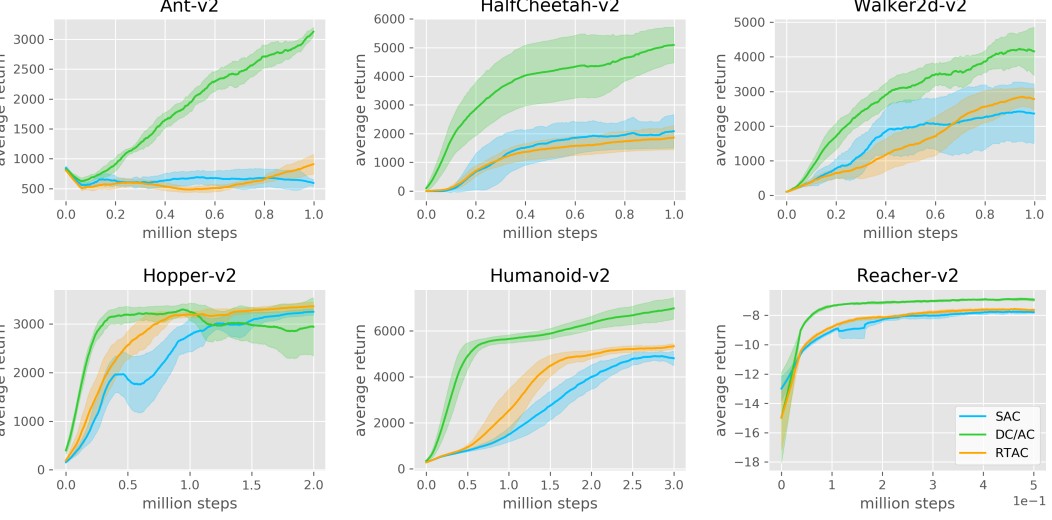

Figure 6: $\omega = 2, \alpha = 3$ (constant delays). With a constant total delay of five time-steps, DCAC exhibits a very strong advantage in performance. All tested algorithms use the same RDMDP augmented observations.

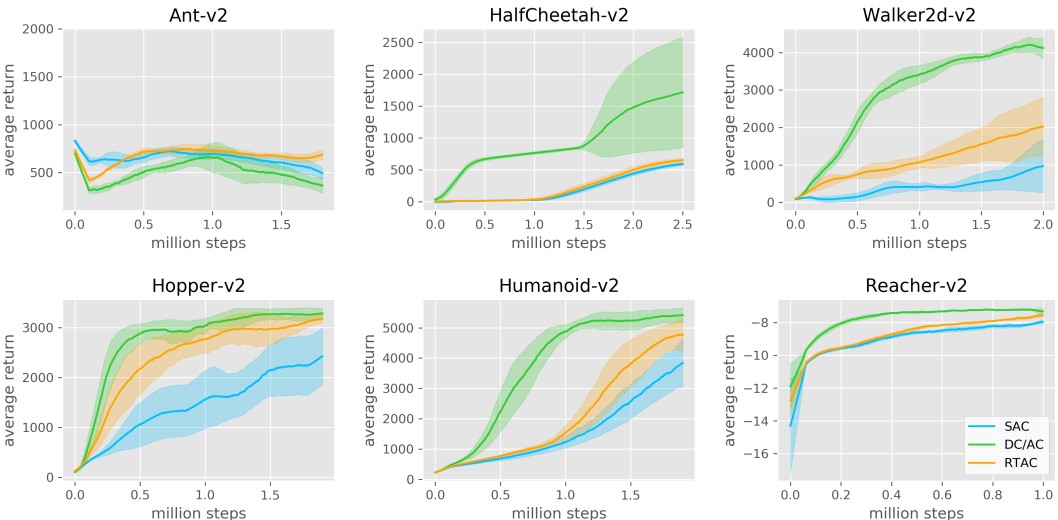

Figure 7: $\alpha, \omega \sim$ WiFi (random delays). DCAC clearly dominates the baselines. Ant became too difficult for all tested algorithms. HalfCheetah also became difficult and only DCAC escapes from local minima.

**Constant delays.** Our first batch of experiments features simple, constantly delayed scenarios. Figure 6 displays the results of the most difficult of these experiments (i.e. where the delays are longest), while the others are provided in Section D.2 of the Appendix. The advantage of using DCAC is obvious in the presence of long constant delays. Note that DCAC reduces to the RTAC (Ramstedt & Pal, 2019) algorithm when $\omega = 0$ and $\alpha = 1$ and behaves as an evolved form of RTAC in the presence of longer constant delays.

**Real-world random delays.** Our second batch of experiments features random delays of different magnitudes. The experiment we chose to present in Figure 7 is motivated by the fact that our approach is designed for real-world applications. Importantly, it provides an example how to implement DCAC in practice (see Appendix A and B for more details). We sample the communication delays for actions and observations from our real-world WiFi dataset, presented in Figure 2. When action or observation communications supersede previous communications, only the most recently produced information is kept. In other words, when an action is received in the undelayed environment, its age is compared to the action that is currently being applied. Then, the one that the agent most recently started to produce is applied. Similarly, when the agent receives a new observation, it only keeps the one that was most recently captured in the undelayed environment (see the right-hand side of Figure 3 for a visual example). We discretize the communication delays by using a time-step of 20ms. Importantly, note that Figure 2 has been cropped to 60ms, but the actual dataset contains outliers that can go as far as 1s. However, long delays (longer than 80ms in our example) are almost always superseded and discarded. Therefore, when such information is received, we clip the corresponding delay with no visible impact in performance: in practice, the maximum acceptable delays are design choices, and can be guided by existing probabilistic timing methods (Santinelli et al., 2017).

## 6 RELATED WORK

We trace our line of research back to Katsikopoulos & Engelbrecht (2003), who provided the first discussion about Delayed Markov Decision Processes. In particular, they were interested in asynchronous rewards, which provides interesting insights in relation to Appendix A.4. Walsh et al. (2008) later re-introduced the notion of "Constantly Delayed Markov Decision Process". While recent advances in deep learning enable implementations of what the authors call an "augmented approach", this was considered intractable at the time because the size of the action buffer grows with the considered delay length. Instead, they studied the case where observations are retrieved with a constant delay and developed a model-based algorithm to predict the current state of the environment. Similarly, Schuitema et al. (2010) developed "memory-less" approaches based on SARSA and vanilla

Q-learning, taking advantage of prior knowledge about the duration of a constant control delay. Hester & Stone (2013) adopted the action buffer-augmented approach to handle random delays, and relied on a decision-tree algorithm to perform credit assignment implicitly. By comparison, our approach relies on delay measurements to perform credit assignment explicitly. More recently, Firoiu et al. (2018) introduced constant action delays to a video game to train agents whose reaction time compares to humans. Similar to previous work, the authors used a state-predictive model, but based on a recurrent neural network architecture. Ramstedt & Pal (2019) formalized the framework of Real-Time Reinforcement Learning (RTRL) that we generalize here to all forms of real-time delays. Initially designed to cope with the fact that inference is not instantaneous in real-world control, the RTRL setting is equivalent to a constantly delayed MDP with $\alpha = 1$ and $\omega = 0$. Finally, Xiao et al. (2020) adopted an alternative approach by considering the influence of the action selection time when action selection is performed within the duration of a larger time-step. However, their framework only allows delays smaller than one time-step, whereas large time-steps are not compatible with high-frequency control.

## 7 Conclusion and future work

We proposed a deep off-policy and planning-free approach that explicitly tackles the credit assignment difficulty introduced by real-world random delays. This is done by taking advantage of delay measurements in order to generate actual on-policy sub-trajectories from off-policy samples. In addition, we provide a theoretical analysis that can easily be reused to derive a wide family of algorithms such as DCAC, whereas previous work mostly dealt with finding approximate ways of modelling the state-space in constantly delayed environments. The action buffer is fundamentally required to define a Markovian state-space for RDMDPs , but it is of course possible to observe this action buffer approximately, e.g. by compressing it in the hidden state of an RNN, which is complementary to our work.

We have designed our approach with real-world applications in mind, and it is easily scalable to a wide variety of scenarios. For practical implementation, see Section 5 and Sections A and B of the Appendix. See also rtgym, a small python helper that we use in future work to easily implement delayed environments in the real world.

To the best of our knowledge, DCAC is the first deep actor-critic approach to exhibit such strong performance on both randomly and constantly delayed settings, as it makes use of the partially known dynamics of the environment to compensate for difficult credit assignment. We believe that our model can be further improved by making use of the fact that our critic estimates the state-value instead of the action-value function. Indeed, in this setting, Ramstedt & Pal (2019) showed that it is possible to simplify the model by merging the actor and the critic networks using the PopArt output normalization (van Hasselt et al., 2016), which we did not try yet and leave for future work.

Our approach handles and adapts to arbitrary choices of time-step duration, although in practice time-steps smaller than the upper bound of the inference time will require a few tricks. We believe that this approach is close to time-step agnostic RL and will investigate this direction in future work.

## Acknowledgments

We thank Pierre-Yves Lajoie, Yoshua Bengio and our anonymous reviewers for their constructive feedback, which greatly helped us improve the article. We also thank ElementAI and Compute Canada for providing the computational resources we used to run our experiments.

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

## A    PRACTICAL CONSIDERATIONS AND SCALABILITY

### A.1    SELF-CORRELATED DELAYS

The separation between $\omega$ and $\alpha$ allows auto-correlated conditional distributions on both delays. This is necessary to allow superseded actions and observations to be discarded. In RDMDPs , the agent keeps the delayed observation that was most recently captured in the undelayed environment. Ideally, it is also ensured by the undelayed environment that the applied action is the action that most recently started being computed by the agent. In practice, this can be ensured by augmenting the actions with timestamps corresponding to the beginning of their computation, and observations with timestamps corresponding to the end of their capture. Thus, the undelayed environment and the agent can keep track of the most recent received timestamp and discard outdated incoming information.

### A.2    HOW TO MEASURE DELAYS

To measure the delays in practice, one possibility is to make use of the aforementioned timestamps. In addition to the augmentations described in A.1, one can augment each observation sent by the undelayed environment with the timestamp of the action that was applied before the end of observation capture. When the agent receives an observation, this observation then contains two timestamps: one that directly corresponds to an action in the buffer (agent's clock), and one that corresponds to when the observation finished being captured (undelayed environment's clock). The identified action in the buffer directly gives the total delay. If the agent and the undelayed environment have e.g. synchronized clocks, the current timestamp minus the timestamp corresponding to observation capture gives the observation delay (and thus we can deduce the action delay).

### A.3    SCALABILITY OF THE ACTION BUFFER

As seen in our WiFi experiment, the maximum delays are design choices in practice. The actual maximum delays can be prohibitively long (e.g. infinite when packets are lost) and would require a long action buffer to be handled in the worst-case scenario. However, in random delays scenarios, long delays are likely to be superseded by shorter delays. Therefore, observations reaching the agent with a total delay that exceeds the chosen $K$ value should simply be discarded, and a procedure implemented to handle the unlikely edge-case where more than $K$ such observations are received in a row. Also note that, although we used a simple action buffer in this work, more clever representations are possible in the presence of long delays, e.g. run-length encoding.

### A.4    DELAYED REWARDS

We have implicitly made a choice when defining the rewards for RDMDPs . Indeed, keep in mind that observations can be dropped (superseded) at the level of the agent. In such cases, we chose to accumulate the rewards corresponding to the lost transitions. When an observation gets repeated because no new observation is available, the corresponding reward is 0, and when a new observation arrives, the corresponding reward contains the sum of intermediate rewards in lost transitions.

In practice, this is ensured for example by making the assumption that the remote robot (i.e. the undelayed environment) can observe its own instantaneous reward. This allows the robot to compute its cumulative reward and send it to the agent along with the observation. The agent can then compute the difference between the last cumulative reward it received from the remote robot and the new one for each incoming observation (NB: outdated observations are discarded so the agent only sees cumulative rewards with time-increasing timestamps).

Alternatively, the practitioner can choose to repeat the delayed rewards along with the repeated delayed observations at the level of the agent (this is what we use to do in earlier versions of the paper). When a trick similar to the aforementioned cannot be implemented, this can be done instead, with no impact on our analysis. However, the reward signal will inherently have a higher variance.

### A.5 LONG OBSERVATION CAPTURE

In practice, it is often the case that observation capture is not instantaneous. In such situation, one should increase the size of the action buffer so that it always includes the actions for which it is unclear whether they have influenced the observation yet or not. Indeed, when observation capture is not instantaneous it is not possible to know which undelayed state(s) it describes. The length of the multi-step backup performed by DCAC doesn't need to be adapted, because it only cares about the first action that is known to not have influenced the delayed observation.

### A.6 COMBINED OBSERVATIONS

Equivalently, if observations are formed of several combined parts that were captured at different times, the action buffer must be long enough to always include the first action that has not influenced the oldest sub-observation yet (i.e. be as long as the maximum possible combined total delay).

## B  IMPLEMENTATION DETAILS

### B.1 MORE INFORMATION AS INPUT TO THE MODEL

The action delay $\alpha$ identifies the action that was applied during the previous time-step. It is needed to define RDMDPs and thus is used by DCAC. However, in practice we can include another piece of information on top of $\alpha$: the delay of the action that is going to be applied in the undelayed environment when the captured observation is sent. We use this additional information as input of the model for all tested algorithms.

### B.2 MODEL ARCHITECTURE

The model we use in all our experiments is composed of two separate multi-layer perceptrons (MLPs): a critic network, and an actor network. Both MLPs are built with the same simple architecture of two hidden layers, 256 units each. The critic outputs a single value, whereas the actor outputs an action distribution with the dimension of the action-space, from which actions are sampled with the reparameterization trick. This architecture is compatible with the second version of SAC described in Haarnoja et al. (2018b). The only difference from the DCAC model is that the SAC critic tracks $q(x)$, and not $v(x)$. Indeed, differently from usual actor-critic algorithms, the output of DCAC's critic approximates the state-value $v(x)$ (instead of the action-value $q(x)$), as it is sufficient to optimize the actor loss described in Definition 7. Weights and biases are initialized with the default Pytorch initializer. Both the actor and the critic are optimized by gradient descent with the Adam optimizer, on losses $L^{\mathrm{DCAC}}(\pi)$ (Equation 10) and $L^{\mathrm{DCAC}}(v)$ (Equation 9), respectively. Classically, we use twin critic networks (Van Hasselt et al., 2015; Fujimoto et al., 2018) with target weight tracking (Mnih et al., 2015) to stabilize training.

### B.3 HYPERPARAMETERS

Other than our neural network architecture, our implementations of SAC, RTAC and DCAC all share the following hyperparameters:

Table 1: Hyperparameters

| Name | Value |
|---|---|
| Optimizer | Adam (Kingma & Ba, 2014) |
| Learning rate | 0.0003 |
| Discount factor ($\gamma$) | 0.99 |
| Batch size | 128 |
| Target weights update coefficient ($\tau$) | 0.005 |
| Gradient steps / environment steps | 1 |
| Reward scale | 5.0 |
| Entropy scale | 1.0 |
| Replay memory size | 1000000 |
| Number of samples before training starts | 10000 |
| Number of critics | 2 |

NB: the target weights are updated according to the following running average: $\bar{\theta} \leftarrow \tau\theta + (1-\tau)\bar{\theta}$

## C  VISUAL EXAMPLES

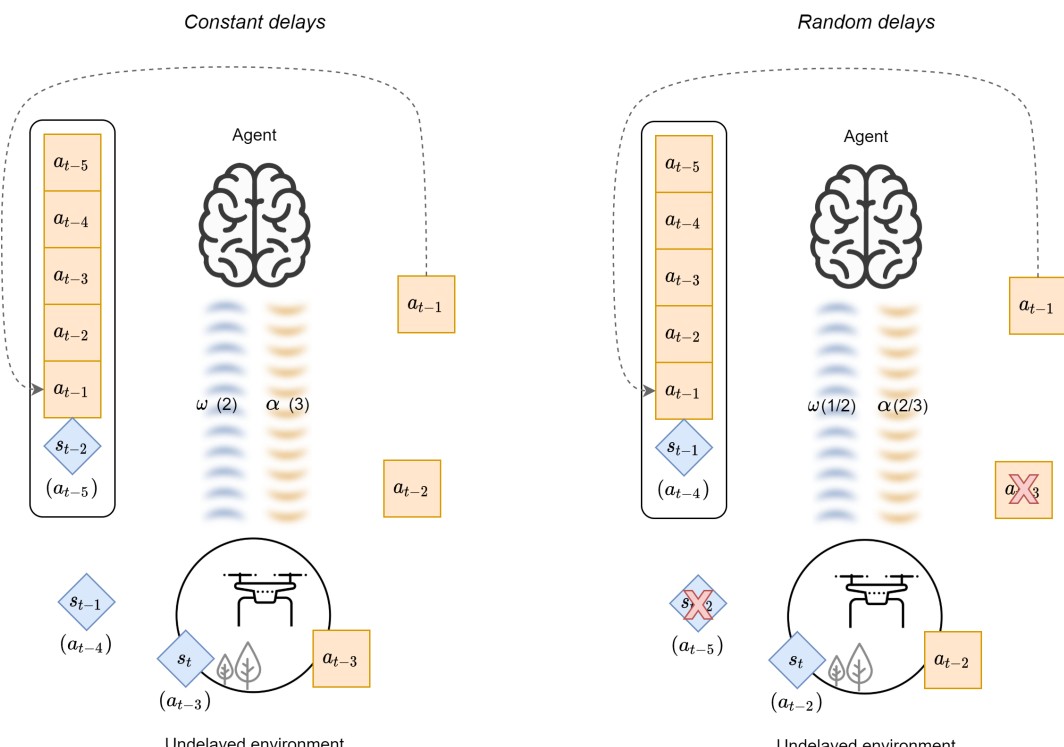

Figure 8: **Left:** Example of Constantly Delayed MDP, with an action delay of three time-steps and an observation delay of two time-steps. Here, actions are indexed by the time at which they started being produced. The augmented observation is composed of an action buffer of the last five computed actions along with the delayed observation $s_{t-2}$. It will be used by the agent to compute action $a_t$. Meanwhile, in the undelayed environment, action $a_{t-3}$ is received and observation $s_t$ is captured. **Right:** Example of Random Delay MDP, with $\alpha \leq 3$ time-steps and $\omega \leq 2$ time-steps. Actions and observations may be superseded due to random delays. In such cases, only the most recently produced actions and observations are kept, the others are discarded (crossed out).

# D ADDITIONAL EXPERIMENTS

## D.1 IMPORTANCE OF THE AUGMENTED OBSERVATION SPACE

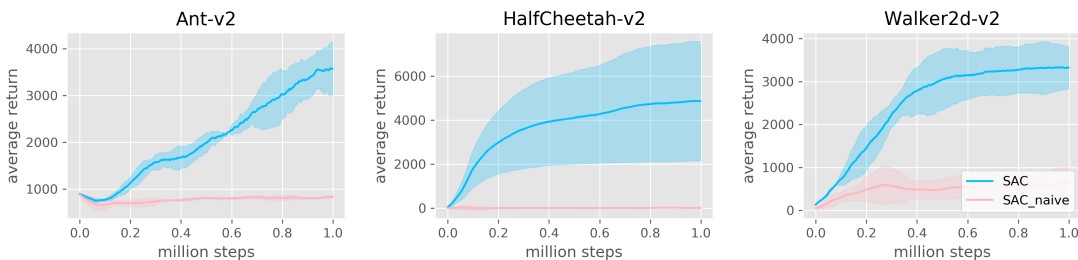

Figure 9: $\omega = 0, \alpha = 1$: We illustrate the importance of the augmented observation space in delayed settings using our simplest task (constant 1-step action delay). Even with this small 1-step constant delay, the delayed observations are not Markov and a naive algorithm using only these observations (here: SAC naive) has near-random results. By comparison, an algorithm using the *RDMDP* augmented observations instead (here: SAC) is able to learn in delayed environments.

## D.2 CONSTANT DELAYS

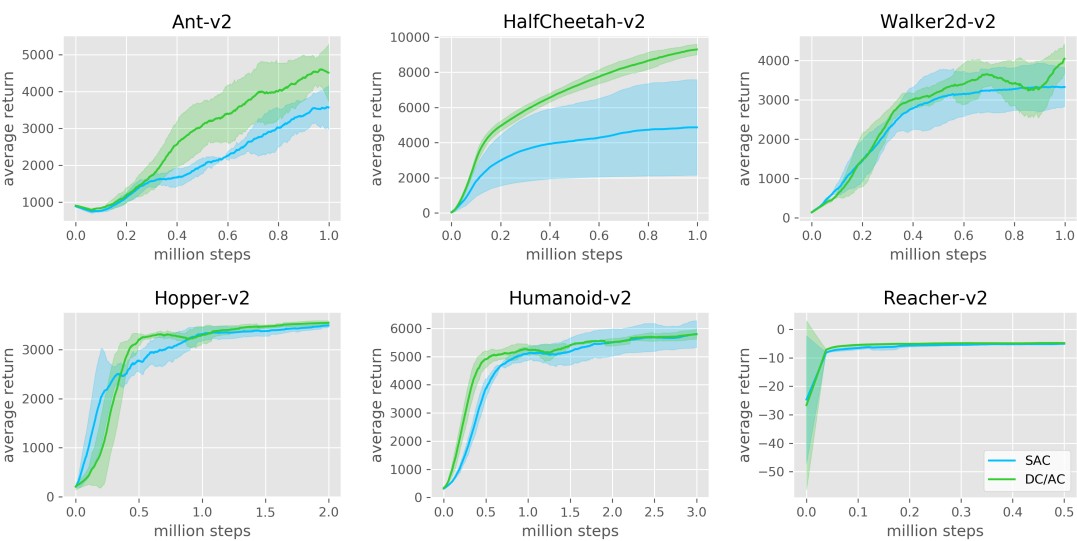

Figure 10: $\omega = 0, \alpha = 1$: This specific setting is equivalent to the RTRL setting (Ramstedt & Pal, 2019), in which DCAC reduces to the vanilla RTAC algorithm (without output normalization and merged networks). DCAC (RTAC) slightly outperforms SAC in this setting.

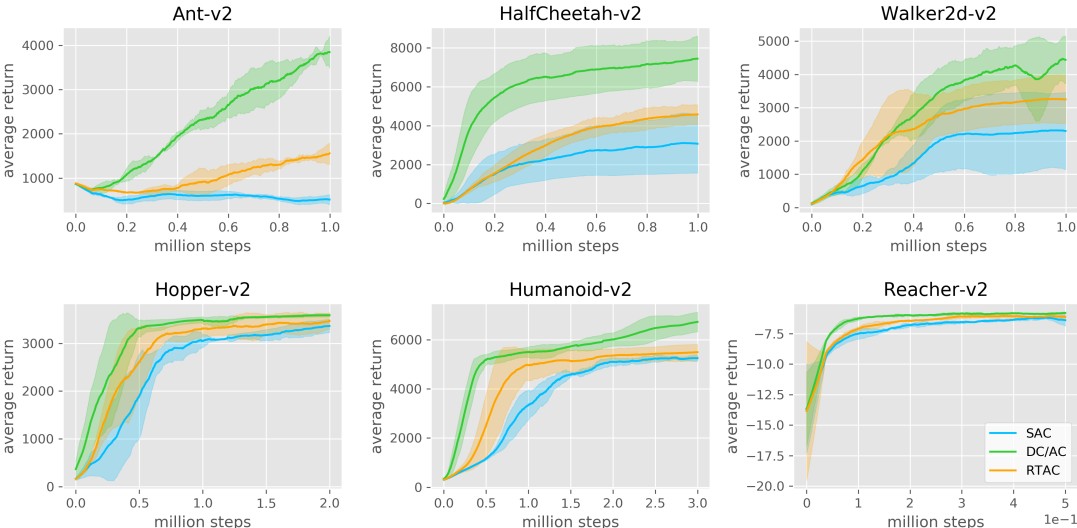

Figure 11: $\omega = 1, \alpha = 2$: In this more difficult setting (total constant delay of 3 instead of 1), DCAC starts really showing its potential, clearly outperforming all other approaches.

## D.3   RANDOM DELAYS

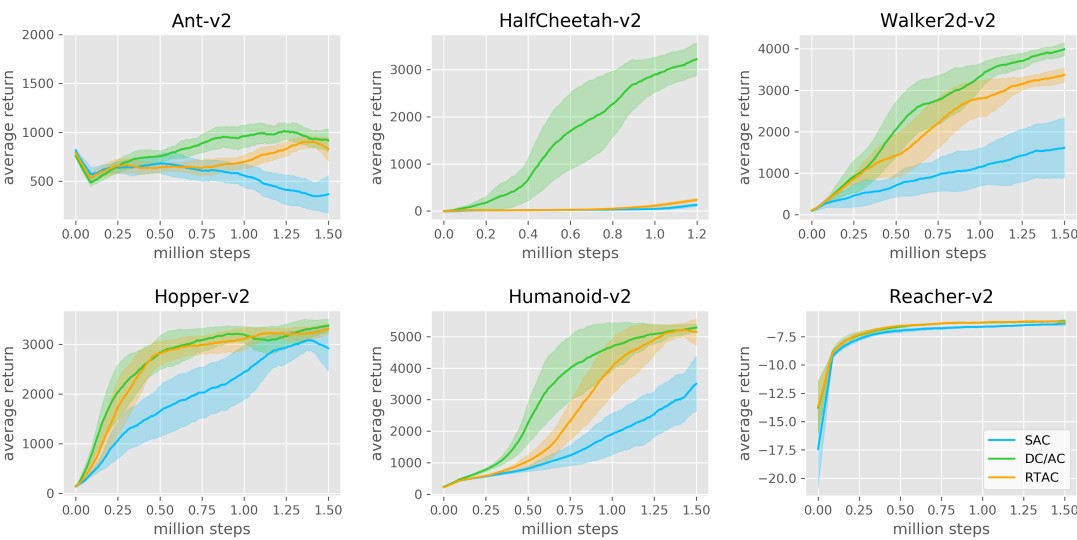

Figure 12: $\omega \in [0; 2], \alpha \in [1; 3]$ (uniformly sampled delays): This experiment is perhaps even more difficult than the WiFi experiment featured in the main paper, because it gives equal probability to all possible delays in the specified ranges (but delays are smaller here which makes it easier for RTAC, because these delays are closer to 1). All tested approaches fail on randomly delayed Ant. For other tasks, the advantage of DCAC is very clear over SAC.

# E    DEFINITIONS

**Definition 8.** The $n$-step state-reward distribution for an environment $E = (S, A, \mu, p)$ and a policy $\pi$ is defined as

$$p_{n+1}^\pi(\underbrace{s', r', \tau_n}_{\tau_{n+1}}|s) = \mathbb{E}_{a\sim\pi(\cdot|s)}[p_n^\pi(\tau_n|s')p(s', r'|s, a)] = \int_A p_n^\pi(\tau_n|s')p(s', r'|s, a)\pi(a|s)da \quad (13)$$

with the base case $p_0^\pi(s) = 1$ and the first iterate $p_1^\pi(s', r'|s) = \int_A p(s', r'|s, a)\pi(a|s)da$.

**Definition 9.** A 1-step action-value estimator is defined as

$$\hat{q}_1(s,a; s',r')=r'+\gamma\, \mathbb{E}_{a'\sim\pi(\cdot|s')}[\hat{q}_0(s',a')]. \quad (14)$$

Part of this estimator is usually another parametric estimator $\hat{q}_0$ (e.g. a neural network trained with stochastic gradient descent).

# F    OTHER MATHEMATICAL RESULTS

## F.1    LEMMA ON STEADY-STATE VALUE ESTIMATION BIAS

**Lemma 3.** The expected bias of the n-step value estimator under the steady-state distribution (if it exists) is

$$\mathbb{E}_{x\sim p_{\text{ss}}^\pi}[\text{bias } \hat{v}_n(x)] = \gamma^n \mathbb{E}_{x\sim p_{\text{ss}}^\pi}[\text{bias } \hat{v}_0(x)] \quad (15)$$

*Proof.* We remind ourselves that the steady state distribution observes

$$p_{\text{ss}}^\pi(x_n) = \mathbb{E}_{x_0\sim p_{\text{ss}}^\pi}[p_n^\pi(..., x_n, r_n|x_0)]. \quad (16)$$

According to Lemma 1 we then have

$$\mathbb{E}_{x_0\sim p_{\text{ss}}^\pi} \text{bias}(\hat{v}_n(x_0, \cdot)) =\gamma^n\mathbb{E}_{...,x_n^*,r_n^*\sim p_n^\pi(\cdot|x_0)}[\text{bias}(\hat{v}_0(x_n^*))] \quad (17)$$

$$=\gamma^n\mathbb{E}_{x\sim p_{\text{ss}}^\pi}[\text{bias } \hat{v}_0(x)]. \quad (18)$$

$\square$

## F.2    LEMMA ON A DIRAC DELTA PRODUCT DISTRIBUTION

**Lemma 4.** For $p(u, v) = \delta(u - c)q(u, v)$ if $q(u, v) < \infty$ for $u = c$ then $p(u, v) = \delta(u - c)q(c, v)$.

*Proof.* If $u = c$ then $p(u, v) = \delta(u - c)q(c, v)$, otherwise $p(u, v) = 0 = \delta(u - c)q(c, v)$    $\square$

## F.3    LEMMA ON F

**Lemma 5.** The dynamics described by $f$ depend neither on the input action nor on a range of actions in the action buffer:

$$f_\Delta(s_1^*, \alpha_1^*, r_1^*|x_0, a_0^\mu) = f_\Delta(s_1^*, \alpha_1^*, r_1^*|x_0^*, a_0^\pi)$$

with $x_0 = s_0, u_0, \omega_0, \alpha_0$ and $x_0^* = s_0^*, u_0^*, \omega_0^*, \alpha_0^*$, given that $s_0, \omega_0, \alpha_0 = s_0^*, \omega_0^*, \alpha_0^*$ and given

$$u_0[\omega_0^* - \delta + \alpha_1^*] = u_0^*[\omega_0^* - \delta + \alpha_1^*] \quad \text{for all} \quad \delta \in \{\Delta, \Delta - 1, \ldots, 0\}$$

*Proof.* We prove by induction.

The base case ($\omega_0^* - \omega_1^* = -1$) is trivial since it does not depend on the inputs that differ.

For the induction step we have

$$f_\Delta(s_1^*, \alpha_1^*, r_1^* | s_0, u_0, \omega_0, \alpha_0, a_0^\mu) =$$
$$\mathbb{E}_{\bar{s}, \bar{\alpha}, \bar{r} \sim f_{\Delta-1}(\cdot | s_0, u_0, \omega_0, \alpha_0, a_0^\mu)}[p(s_1^*, r_1^* - \bar{r} | \bar{s}, u[\omega_0 - \Delta + \alpha_1^*]) \, p_\alpha(\alpha_1^* | \bar{\alpha})] \quad (19)$$

Because of our condition on $u_0$ and $u_0^*$ and the fact that $\omega_0 = \omega_0^*$ this is equal to

$$\mathbb{E}_{\bar{s}, \bar{\alpha}, \bar{r} \sim f_{\Delta-1}(\cdot | s_0, u_0, \omega_0, \alpha_0, a_0^\mu)}[p(s_1^*, r_1^* - \bar{r} | \bar{s}, u_0^*[\omega_0^* - \Delta + \alpha_1^*]) \, p_\alpha(\alpha_1^* | \bar{\alpha})]$$

We can now use the induction hypothesis since the conditions on $s_0, u_0, \omega_0, \alpha_0$ are still met when $\Delta \leftarrow \Delta - 1$.

$$\mathbb{E}_{\bar{s}, \bar{\alpha}, \bar{r} \sim f_{\Delta-1}(\cdot | s_0^*, u_0^*, \omega_0^*, \alpha_0^*, a_0^\pi)}[p(s_1^*, r_1^* - \bar{r} | \bar{s}, u_0^*[\omega_0^* - \Delta + \alpha_1^*]) \, p_\alpha(\alpha_1^* | \bar{\alpha})]$$
$$= f_\Delta(s_1^*, \alpha_1^*, r_1^* | x_0^*, a_0^\pi) \quad (20)$$

$\square$

### F.4 LEMMA ON PARTIAL RESAMPLING

**Lemma 6.** Partially resampling trajectories collected under a policy $\mu$ according to $\sigma_n^\pi$ transforms them into trajectories distributed according to $\pi$.

$$\mathbb{E}_{\tau_n \sim p_n^\mu(\cdot | x_0)}[\sigma_n^\pi(\tau_n^* | x_0^*; \tau_n)] = p_n^\pi(\tau_n^* | x_0^*)$$

with $x_0 = s_0, u_0, \omega_0, \alpha_0$ and $x_0^* = s_0^*, u_0^*, \omega_0^*, \alpha_0^*$, on the condition that $s_0, \omega_0, \alpha_0 = s_0^*, \omega_0^*, \alpha_0^*$ and on the condition that the actions in the initial action buffers $u_0$ and $u_0^*$ that are applied in the following trajectory are the same, i.e.

$$u_0[k : \text{end}] = u_0^*[k : \text{end}] \quad \text{with} \quad k = \min_i(\omega_{i+1}^* + \alpha_{i+1}^* - i) \quad \text{for} \quad i \in \{0, n-1\}$$

and for the trajectory $\tau_n^* = (s_1^*, u_1^*, \omega_1^*, \alpha_1^*, \ldots, s_n^*, u_n^*, \omega_n^*, \alpha_n^*)$.

*Proof.* We start with the induction base for $n = 0$. The theorem is trivial in this case since we have 0-length trajectories () and $p_0^\mu(() | x_0) = \sigma_0^\pi(() | x_0^*; ()) = p_0^\pi(() | x_0^*) = 1$.

For the induction step we start with the left hand side of the lemma's main equation.

$$\mathbb{E}_{\tau_n \sim p_n^\mu(\cdot | x_0)}[\sigma_n^\pi(\tau_n^* | x_0^*; \tau_n)]$$
$$= \mathbb{E}_{a_0^\mu \sim \mu(\cdot | x_0)}[\mathbb{E}_{x_1, r_1 \sim \tilde{p}(x_1, r_1 | x_0, a_0^\mu)}[\mathbb{E}_{\tau_{n-1} \sim p_{n-1}^\mu(\cdot | x_1)}[\sigma_n^\pi(\tau_n^* | x_0^*; x_1, r_1, \tau_{n-1})]]] \quad (21)$$

with

$$\tilde{p}(s_1, u_1, \omega_1, \alpha_1, r_1 | s_0, u_0, \omega_0, \alpha_0, a_0^\mu) = f_{\omega_0 - \omega_1}(s_1, \alpha_1, r_1 | s_0, u_0, \omega_0, \alpha_0, a_0^\mu) \, p_\omega(\omega_1 | \omega_0) \, p_u(u_1 | u_0, a_0^\mu)$$

Plugging that and solving the integral over $u_1$ yields

$$= \mathbb{E}_{a_0^\mu \sim \mu(\cdot | x_0)}[\mathbb{E}_{\omega_1 \sim p_\omega(\cdot | \omega_0)}[\mathbb{E}_{s_1, \alpha_1, r_1 \sim f_{\omega_0 - \omega_1}(\cdot | s_0, u_0, \omega_0, \alpha_0, a_0^\mu)}[$$
$$\mathbb{E}_{\tau_{n-1} \sim p_{n-1}^\mu(\cdot | s_1, (a_0^\mu, u_0[1:-1]), \omega_1, \alpha_1)}[\sigma_n^\pi(\tau_n^* | x_0^*; s_1, (a^\mu, u_0[1:-1]), \omega_1, \alpha_1, r_1, \tau_{n-1})]]]] \quad (22)$$

Rolling out $\sigma_n^\pi$ by one step and integrating out $s_1, \omega_1, \alpha_1, r_1$ yields

$$= \mathbb{E}_{a_0^\mu \sim \mu(\cdot|x_0)}[\mathbb{E}_{\tau_{n-1} \sim p_{n-1}^\mu(\cdot|s_1^*,(a_0^\mu,u_0[1:-1]),\omega_1^*,\alpha_1^*)}[\mathbb{E}_{a_0^\pi \sim \pi(\cdot|x_0^*)}[\delta(u_1^* - (a_0^\pi, u_0^*[1:-1]))$$
$$\sigma_{n-1}^\pi(\tau_{n-1}^*|s_1^*, u_1^*, \omega_1^*, \alpha_1^*; \tau_{n-1})f_{\omega_0-\omega_1^*}(s_1^*, \alpha_1^*, r_1^*|s_0, u_0, \omega_0, \alpha_0, a_0^\mu) \, p_\omega(\omega_1^*|\omega_0)]]] \quad (23)$$

Reordering terms and substituting $s_0, \omega_0, \alpha_0 = s_0^*, \omega_0^*, \alpha_0^*$ yields

$$= p_\omega(\omega_1^*|\omega_0^*)\mathbb{E}_{a_0^\pi \sim \pi(\cdot|x_0^*)}[\delta(u_1^* - (a_0^\pi, u_0^*[1:-1]))$$
$$\mathbb{E}_{a_0^\mu \sim \mu(\cdot|x_0)}[f_{\omega_0^*-\omega_1^*}(s_1^*, \alpha_1^*, r_1^*|x_0, a_0^\mu)$$
$$\mathbb{E}_{\tau_{n-1} \sim p_{n-1}^\mu(\cdot|s_1^*,(a^\mu, u_0[1:-1]),\omega_1^*,\alpha_1^*)}[\sigma_{n-1}^\pi(\tau_{n-1}^*|x_1^*; \tau_{n-1})]]] \quad (24)$$

We can substitute the $f$ term according to Lemma 5 since the condition between $x_0$ and $x_0^*$ is met. More precisely the condition on $u_0$ and $u_0^*$ is met because $k \leq \omega_0^* - \Delta + \alpha_1^* = \omega_1^* + \alpha_1^*$. After the substitution we have

$$= p_\omega(\omega_1^*|\omega_0^*)\mathbb{E}_{a_0^\pi \sim \pi(\cdot|x_0^*)}[\delta(u_1^* - (a_0^\pi, u_0^*[1:-1])) \, f_{\omega_0^*-\omega_1^*}(s_1^*, \alpha_1^*, r_1^*|x_0^*, a_0^\pi)$$
$$\mathbb{E}_{a_0^\mu \sim \mu(\cdot|x_0)}[\mathbb{E}_{\tau_{n-1} \sim p_{n-1}^\mu(\cdot|s_1^*,(a^\mu,u_0[1:-1]),\omega_1^*,\alpha_1^*)}[\sigma_{n-1}^\pi(\tau_{n-1}^*|x_1^*; \tau_{n-1})]]] \quad (25)$$

We can substitute the induction hypothesis in the following form.

$$\mathbb{E}_{\tau_{n-1} \sim p_{n-1}^\mu(\cdot|x_1)}[\sigma_{n-1}^\pi(\tau_{n-1}^*|x_1^*; \tau_{n-1})] = p_{n-1}^\pi(\tau_{n-1}^*|x_1^*)$$

on the condition that

$$u_1[k : \text{end}] = u_1^*[k : \text{end}] \quad \text{with} \quad k = \min_i(\omega_{i+2}^* + \alpha_{i+2}^* - i) \quad \text{for} \quad i \in \{0, n-2\}$$

for the trajectory $\tau_{n-1}^* = (s_2^*, u_2^*, \omega_2^*, \alpha_2^*, \ldots, s_n^*, u_n^*, \omega_n^*, \alpha_n^*)$. To check that this condition is met we observe that $u_1 = (a_0^\mu, u_0[1:-1])$ and substitute $u_1^* = (a_0^\pi, u_0^*[1:-1])$ (made possible by Lemma 4) which means that

$$u_0[k-1 : \text{end}] = u_0^*[k-1 : \text{end}] \quad \text{with} \quad k = \min_i(\omega_{i+2}^* + \alpha_{i+2}^* - i) \quad \text{for} \quad i \in \{0, n-2\}$$

Substituting the induction hypothesis yields

$$= p_\omega(\omega_1^*|\omega_0^*)\mathbb{E}_{a_0^\pi \sim \pi(\cdot|x_0^*)}[\delta(u_1^* - (a_0^\pi, u_0[1:-1])) \, f_{\omega_0^*-\omega_1^*}(s_1^*, \alpha_1^*, r_1^*|x_0^*, a_0^\pi) \, p_{n-1}^\pi(\tau_{n-1}^*|x_1^*)]$$
$$(26)$$

which is

$$\mathbb{E}_{a_0^\pi \sim \pi(\cdot|x_0^*)}[p_{n-1}^\pi(\tau_{n-1}^*|x_1^*) \, \tilde{p}(x_1^*, r_1^*|x_0^*, a_0^\pi)] = p_n^\pi(\tau_n^*|x_0^*)$$

$$\square$$

# G  PROOFS OF THE RESULTS FROM THE MAIN PAPER

## A

**Theorem 1.** The partial trajectory resampling operator $\sigma_n^\pi$ (Def. 3) transforms off-policy trajectories into on-policy trajectories

$$\mathbb{E}_{\tau_n \sim p_n^\mu(\cdot|x_0)}[\sigma_n^\pi(\tau_n^*|x_0; \tau_n)] = p_n^\pi(\tau_n^*|x_0) \quad (3)$$

on the condition that none of the delayed observations depend on any of the resampled actions, i.e.

$$\omega_t^* + \alpha_t^* \geq t \tag{4}$$

where $t$ indexes the trajectory $\tau_n^* = (s_1^*, u_1^*, \omega_1^*, \alpha_1^*, r_1^*, \ldots, s_n^*, u_n^*, \omega_n^*, \alpha_n^*, r_n^*)$ from 1 to $n$.

*Proof.* The theorem is a special case of Lemma 6 with $x_0 = x_0^*$. This allows us to simplify the condition in the lemma as we show next.

Since $u_0 = u_0^*$ we can allow all $k \geq 1$ which is the minimum allowed index for $u$. Therefore we must ensure $1 \leq \min_i(\omega_{i+1}^* + \alpha_{i+1}^* - i)$. Since the $\min$ must be larger than 1 then all arguments must be larger than 1 which means this is equivalent to

$$1 \leq \omega_{i+1}^* + \alpha_{i+1}^* - i \quad \text{for} \quad i \in \{0, n-1\}.$$

This can be transformed into

$$\omega_t^* + \alpha_t^* \geq t \quad \text{for} \quad t \in \{1, n\} \tag{27}$$

$\square$

**Lemma 1.** The n-step value estimator has the following bias:

$$\text{bias}(\hat{v}_n(x_0, \cdot)) = \gamma^n \mathbb{E}_{\ldots, x_n^*, r_n^* \sim p_n^\pi(\cdot|x_0)}[\text{bias}(\hat{v}_0(x_n^*))] \tag{6}$$

*Proof.*

$$\text{bias}(\hat{v}_n(x_0, \cdot)) = \mathbb{E}_{\tau_n^* \sim p_n^\pi(\cdot|x_0)}[\hat{v}_n(x_0, \tau_n^*) - v^\pi(x_0)] \tag{28}$$

$$= \mathbb{E}_{\tau_n^* \sim p_n^\pi(\cdot|x_0)}[r_1^* + \gamma \hat{v}_{n-1}(x_1^*; \tau_{n-1}^*)] - \mathbb{E}_{a_0 \sim \pi(\cdot|x_0)}[\mathbb{E}_{r_1^*, x_1^* \sim \tilde{p}(\cdot|x_0, a_0)}[r_1^* + \gamma v^\pi(x_1^*)]] \tag{29}$$

$$= \mathbb{E}_{\tau_n^* \sim p_n^\pi(\cdot|x_0)}[r_1^* + \gamma \hat{v}_{n-1}(x_1^*; \tau_{n-1}^*) - r_1^* - \gamma v^\pi(x_1^*)] \tag{30}$$

$$= \gamma \mathbb{E}_{\tau_n^* \sim p_n^\pi(\cdot|x_0)}[\hat{v}_{n-1}(x_1^*; \tau_{n-1}^*) - v^\pi(x_1^*)] \tag{31}$$

$$= \ldots \tag{32}$$

$$= \gamma^n \mathbb{E}_{\tau_n^* \sim p_n^\pi(\cdot|x_0)}[\hat{v}_0(x_n^*) - v^\pi(x_n^*)] \tag{33}$$

$$= \gamma^n \mathbb{E}_{\ldots, x_n^*, r_n^* \sim p_n^\pi(\cdot|x_0)}[\text{bias}(\hat{v}_0(x_n^*))] \tag{34}$$

$\square$

# B

**Lemma 2.** In a RDMDP $(E, p_\omega, p_\alpha)$ the soft value function is:

$$v^{\text{soft}}(x_0^*) = \mathbb{E}_{a \sim \pi(\cdot|x_0^*)}[\mathbb{E}_{x_1^*, r_1^* \sim \tilde{p}(\cdot|x_0^*, a)}[r_1^* + \gamma v^{\text{soft}}(x_1^*)] - \log \pi(a|x_0^*)] \tag{7}$$

*Proof.* The soft value function for an environment $(X, A, \bar{\mu}, \bar{p})$ is defined as

$$v^{\text{soft}}(x_0^*) = \mathbb{E}_{a \sim \pi(\cdot|x_0^*)}[q^{\text{soft}}(x_0^*, a) - \log \pi(a|x_0^*)] \tag{35}$$

where

$$q^{\text{soft}}(x_0^*, a) = \mathbb{E}_{x_1^*, r_1^* \sim \bar{p}(x_0^*, a)}[r_1^* + \gamma v^{\text{soft}}(x_1^*)] \tag{36}$$

If $(X, A, \bar{\mu}, \bar{p}) = \text{RDMDP}(E, p_\omega, p_\alpha) = (X, A, \tilde{\mu}, \tilde{p})$ with $E = (S, A, \mu, p)$ this is

$$q^{\text{soft}}(x_0^*, a) = \mathbb{E}_{x_1^*, r_1^* \sim \tilde{p}(\cdot|x_0^*, a)}[r_1^* + \gamma v^{\text{soft}}(x_1^*)] \tag{37}$$

and

$$v^{\text{soft}}(x_0^*) = \mathbb{E}_{a \sim \pi(\cdot|x_0^*)}[\mathbb{E}_{x_1^*, r_1^* \sim \tilde{p}(\cdot|x_0^*)}[r_1^* + \gamma v^{\text{soft}}(x_1^*)] - \log \pi(a|x_0^*)] \tag{38}$$

$\square$

**Proposition 1.** The DCAC actor loss is a less biased version of the SAC actor loss with

$$\mathrm{bias}(L_\pi^{\mathrm{DCAC}}) = \mathbb{E}_n[\gamma^n]\,\mathrm{bias}(L_\pi^{\mathrm{SAC}}) \tag{11}$$

assuming both are using similarly biased parametric value estimators to compute the loss, i.e.

$$\mathrm{bias}(\hat{v}_0^{\mathrm{soft}}(x)) = \mathbb{E}_{a\sim\pi(\cdot|x)}[\mathrm{bias}(\hat{q}_0^{\mathrm{soft}}(x,a))] \tag{12}$$

*Proof.* Note that for simplicity, we also assume that the states in the replay memory are distributed according to the steady-state distribution, i.e. $D \sim p_{\mathrm{ss}}^\pi$. This assumption could be avoided by making more complicated assumptions about the biases of the state-value and action-value estimators.

We now start with the bias of the DCAC loss with respect to an unbiased SAC loss using the true action-value function,

$$\mathrm{bias}(L_\pi^{\mathrm{DCAC}}) = L_\pi^{\mathrm{DCAC}} - L_\pi^{\mathrm{SAC\,-UB}} \tag{39}$$

where

$$L_\pi^{\mathrm{DCAC}} = -\,\mathbb{E}_{x_0,\tau_n\sim\mathcal{D}}\,\mathbb{E}_{\tau_n^*\sim\sigma_n^\pi(\cdot|x_0;\tau_n)}[\hat{v}_n^{\mathrm{soft}}(x_0;\,\tau_n^*)] \tag{40}$$

$$= -\,\mathbb{E}_{x_0\sim D}\mathbb{E}_n\mathbb{E}_{\tau_n^*\sim p_n^\pi(\cdot|x_0)}[\hat{v}_n^{\mathrm{soft}}(x_0;\,\tau_n^*)] \quad | \text{ Theorem 1} \tag{41}$$

and

$$L_\pi^{\mathrm{SAC\,-UB}} = \mathbb{E}_{x_0\sim\mathcal{D}}[\mathbb{E}_{a\sim\pi(\cdot|x_0)}[\log\pi(a|x_0) - q^{\mathrm{soft}}(x_0,a)]] \tag{42}$$

$$= \mathbb{E}_{x_0\sim\mathcal{D}}[v^{\mathrm{soft}}(x_0)]. \tag{43}$$

Substituting these we have

$$\mathrm{bias}(L_\pi^{\mathrm{DCAC}}) = \mathbb{E}_{x_0\sim D}\mathbb{E}_n[\hat{v}_n^{\mathrm{soft}}(x_0;\,\tau_n^*) - v^{\mathrm{soft}}(x_0)] \tag{44}$$

$$= \mathbb{E}_{x_0\sim D}\mathbb{E}_n[\mathrm{bias}(\hat{v}_n^{\mathrm{soft}}(x_0;\cdot)] \tag{45}$$

$$= \mathbb{E}_{x_0\sim D}\mathbb{E}_n[\gamma^n\mathbb{E}_{\ldots,x_n,r_n\sim p_n^\pi(\cdot|x_0)}[\mathrm{bias}(\hat{v}_0^{\mathrm{soft}}(x_n))]] \quad | \text{ Lemma 1} \tag{46}$$

$$= \mathbb{E}_n[\gamma^n]\,\mathbb{E}_{x\sim D}[\mathrm{bias}(\hat{v}_0^{\mathrm{soft}}(x))] \quad | \text{ using } D \sim p_{\mathrm{ss}}^\pi \text{ and Lemma 3} \tag{47}$$

$$= \mathbb{E}_n[\gamma^n]\,\mathbb{E}_{x\sim D}[\mathbb{E}_{a\sim\pi(\cdot|x)}[\mathrm{bias}(\hat{q}_0^{\mathrm{soft}}(x,a))]] \quad | \text{ Equation 12} \tag{48}$$

$$= \mathbb{E}_n[\gamma^n]\,\mathbb{E}_{x\sim D}[\mathbb{E}_{a\sim\pi(\cdot|x)}[\hat{q}_0^{\mathrm{soft}}(x,a) - q^{\mathrm{soft}}(x,a)]] \tag{49}$$

$$= \mathbb{E}_n[\gamma^n]\,(L_\pi^{\mathrm{SAC}} - L_\pi^{\mathrm{SAC\,-UB}}) \tag{50}$$

$$= \mathbb{E}_n[\gamma^n]\,\mathrm{bias}(L_\pi^{\mathrm{SAC}}) \tag{51}$$

$$\square$$

