# OpenReview forum: "Reinforcement Learning with Random Delays"
_ICLR.cc/2021/Conference — ICLR 2021 Poster_

### Official Review · AnonReviewer2 · 2020-10-26
**Clear motivation but confusing contribution**

**Rating:** 3
**Confidence:** 3

**Review:**

This work tackles an existing phenomenon that is often ignored in real-world control problems -- stochastic-lengthed delays. While the motivation and examples are clear, I feel I'm completely baffled by the theory that follows.

I find the definitions and, consequently, the results, extremely hard to follow. Namely, in Def. 1 — why is the state-space a product of $\mathbb{R}^2$? Is this due to the action and observation delay values? If so, why are they continuous and not discrete (as mentioned in the paper)?

Then, in the same definition, comes the most confusing equation regarding $f_\Delta$. What exactly is it? How can it be part of a transition probability while according to Def. 2 it is an expectation? Also, what is $s^*$ there and why is $r’-r^*$ a relevant term? And most puzzling to me is the fact that $f_\Delta$ itself is recursive. That is new and surprising but barely receives any attention in the text and gets me wondering what does that imply on the process and algorithms.

Similar to the confusing definitions, the text itself is very hard to comprehend as well. For example, one paragraph before Sec. 3.1 and the first one discuss an off-policy partial trajectory resampling method using very vague arguments, and the relation to what was presented up to that point in the paper is loose. Honestly, I read those two paragraphs a couple of times and couldn't understand them. Then, the following definition and theorem 1 that follow are as confusing to me as the text. At that point, I felt I could not follow the paper anymore.

Lastly, the experiments apparently exhibit good results, but I cannot say anything smarter on it. Since I couldn’t understand the analysis that preceded the algorithm, I cannot appreciate its qualities.


Additional comments:
1. The literature review is very scarce. Two examples of prior art dealing directly with RL with stochastic delays are [1, 2]. Additional multiple recent citations using SOTA algorithms for constant delay are also missing.
2. Often, unclear sentences are either not backed up by references (e.g., in Sec. 2.1, “it is also possible to do much better when the delays themselves are also part of the state-space”), or when the reader is referred to the appendix, but there, no compelling argument to the original claim is found (e.g. Sec. 2.1,  $r’-r^*$ explanation with reference to B.2).

[1] Katsikopoulos, K. V., & Engelbrecht, S. E. (2003). Markov decision processes with delays and asynchronous cost collection. IEEE transactions on automatic control, 48(4), 568-574.

[2] Campbell, J. S., Givigi, S. N., & Schwartz, H. M. (2016). Multiple model Q-learning for stochastic asynchronous rewards. Journal of Intelligent & Robotic Systems, 81(3-4), 407-422.

---

> ### Author Response · Authors · 2020-11-15
> **Thank you for your review. Please reassess our paper and evaluate it based on our main contribution**
>
> Thank you for reviewing our paper. The core idea of our approach is actually simple: We can arbitrarily re-write the actions that haven't come into play yet, so we can do n-step returns without bias where $n$ is the time it takes the first on-policy action to have an effect in the environment. However, describing this in a mathematically accurate way is fairly involved. We tried our best to make this as clear as possible.
>
> Unfortunately, you seem to have disregarded this core idea which is our main contribution. All other reviewers were able to understand this despite having some problems with the complexities of certain Definitions.
>
> Your comments are almost all about the mathematical description of RDMDPs in Definition 1 and 2. While we appreciate your careful analysis of this part of our paper, this is actually not where our main contribution lies. In fact, precise understanding of Definition 1 and 2 aren't even required to understand our main contribution (except the proofs).
>
> We ask you to please reassess our paper and evaluate it based on our main contribution.
>
> Of course we also want to address your detailed comments and questions about Definition 1 and 2 which we will be doing now:
>
> We believe Definition 1 is hard to follow because it is a partially observable environment described as a MDP. We described it as a MDP because it is the standard for reinforcement learning and nice to work with.
>
> > in Def. 1 — why is the state-space a product of $\mathbb{R}^2$? Is this due to the action and observation delay values? If so, why are they continuous and not discrete (as mentioned in the paper)?
>
> You are right, we changed the $\mathbb{R}^2$ to $\mathbb{N}^2$.
>
> > Then, in the same definition, comes the most confusing equation regarding $f_\Delta$. What exactly is it?
>
> $f_\Delta$ is describing what is happening in the undelayed environment. It is necessary to make it recursive because the agent seeing the undelayed environment through delayed observations can see the observations advance multiple steps at once (or none at all).
>
> >How can it be part of a transition probability while according to Def. 2 it is an expectation?
>
> $f_\Delta$ is a marginal distribution which are always defined as expectations.
>
> >Also what is $s^*$ there
>
> This is an intermediate observation that the agent will never see due to the sequence of delays (which is why it is marginalized over). This happens whenever observation delays are shrinking from one timestep to the next. The observations that haven't been seen by the agent but that aren't the most recent observation will be skipped.
>
> > why is $r'-r^*$ a relevant term?
>
> Similar to how $s^*$ describes skipped observations, $r^*$ describes skipped rewards here. Except that we don't exactly skip them. Instead we sum all of the skipped rewards. This sum of random rewards expresses itself as a convolution when represented as a probability density.
>
>
> > Two examples of prior art dealing directly with RL with stochastic delays are [1, 2].
>
> Reference 1 is relevant and has been added to the literature review. Reference 2 is not relevant to our paper. It doesn't deal with delays except delays in the reward signal which is not something our paper is concerned with. Reward delays are trivial as one can simply undelay these rewards during training.

---

### Official Review · AnonReviewer1 · 2020-10-27
**good results but needs clearer algo description and some missing related work**

**Rating:** 6
**Confidence:** 4

**Review:**

Summary:
The paper introduces an algorithm for the case where actions have delayed effects in RL, and specifically in the case where the delay is random.  A resampling approach is applied to off policy buffered data in order to align it with the current policy and this approach is integrated into a SAC architecture, creating the new DCAC algorithm.  Empirical results in constant-delay and random-delay environments show the algorithm outperforming baselines.

Review:
The strongest part of the paper are the empirical results, which clearly show the approach has merit over an augmented solution and an algorithm built for one step delays.  The approach is also strong because it deals with random delays which are not as well studied in the literature (though an older related work was missed there, see below).

The weakness of the paper is the lack of clarity in the description of the Algorithm itself and particularly in the crucial resampling operator in Definition 3, which is very unclear and makes the solution irreproducible with the current description.  In comparison to related work, the authors claim they are the first to deal with random delays but this is incorrect – Texplore (Hester and Stone) dealt with this situation and there is older work (cited below) that also addressed this problem.  In addition, an empirical comparison to Firoiu et al.’s approach, which is the state of the art in Deep RL for constant delayed observations is needed in the first set of experiments.

Detailed Notes:
The heart of the paper is Section 3.1, which describes the overall algorithm, but unfortunately this section is very difficult to understand.  The confusion begins in the first paragraph of section 3.1 on the bottom of page 3.  The discussions of actions that do not affect observations is too high level and it is unclear what situation exactly the authors are referring to.  A concrete example, even in a simple domain, could make the author’s point clear – can you provide such an example?

Definition 3, the resampling operation, is the most important part of the paper, but it is unclear what is actually happening here.  Is sigma returning a probability of replacing one term with another, or is it a mapping?  What exactly is the delta term doing or is it a function?  And more conceptually, how does this replacement work if two policies don’t share actions at a state?  If the policy that generated in the data in the buffer is “always do action 1” and the new policy is “always do action 2”, resampling is impossible right?  The whole procedure should really be written as pseudocode to make the algorithm reproducible and analyzable.

I found the motivation for the resampling also hard to follow.  Why is on-policy data even needed here?  SAC, as I understand it, is an off-policy algorithm so it should be able to handle off policy samples.  And the delayed observations are the result of the actual actions taken, independently of the policy that produced them.  So can’t one compute the estimated states from the stored actions and provide that data to SAC?  Why is resampling even needed?

Relation to existing work:
Firoiu et al.’s approach should appear in the empirical comparisons on constant-delayed MDPs.  That work is the state of the art on Deep RL with constant delays and while the new algorithm deals with random delays, since there is a constant-delay testbed described here, a comparison is warranted, especially because the overmatched baselines are similar between the current work and Firoiu et al.’s

Two references that were missed but seem highly relevant since they dealt with random delays:
TEXPLORE: real-time sample-efficient reinforcement learning for robots (Hester and Stone) – builds a decision tree representation of delay effects to generalize and deal with non-constant delays.  This solution should be compared to in some way.
“Markov decision processes with delays and asynchronous cost collection” by Katsikopoulos, & Engelbrecht (2003) was the first paper to study random (non-constant) delays in MDPs and should be correctly cited here as the seminal work.

Finally, the description of the prior work in the introduction as “proposing on-policy planning algorithms” does not make sense.  First, planning algorithms by their definition are off-policy since they create new policies.  Secondly, several of the approaches in the referenced works, including Walsh et al. use off-policy algorithms, so that characterization seems incorrect.

Minor notes:
Page 3 – ‘their performance will still deteriorate because of the more difficult credit assignment’ – that is only part of the problem.  The augmented space also causes significant exploration and generalization burdens.
Page 3 - `One solution is to perform on-policy multi step rollouts’ – I don’t see this as a “solution”.  It will certainly generate helpful information, but simply doing rollouts is not an algorithm.  What do you do with them?  How do you combine them?  This is not a good point of comparison because it is not a full procedure.
Page 5, second line – the formed the sub trajectory (fix one of ‘the’ instances)

---

> ### Author Response · Authors · 2020-11-14
> **Thank you for your detailed feedback**
>
> We thank you for the amount of time spent on reviewing our paper, and for your very detailed feedback. We address most of your concerns thereafter.
>
> > The weakness of the paper is the lack of clarity in the description of the Algorithm itself and particularly in the crucial resampling operator in Definition 3. Section 3.1 is very difficult to understand.
>
> We have added an explanatory figure for Definition 3 which should make it easier to understand, and we have released a fairly commented implementation of DCAC (c.f. the general post). We also reformulated a few sentences in Section 3.1 in order to make it easier to follow.
>
> >Definition3: Is sigma returning a probability of replacing one term with another, or is it a mapping? What exactly is the delta term doing or is it a function?
>
> Sigma is defined as a conditional probability density in Definition 3, but in practice one can see it as a mapping because the probability of replacing one term with another is 1 when marginalized. $\delta(a|b)$ stands for the delta Dirac distribution (c.f. Section 2.1). When marginalized, it sets a=b with probability 1 (one can simply see it as assigning $b$ to $a$).
>
>
> >And more conceptually, how does this replacement work if two policies don’t share actions at a state? If the policy that generated in the data in the buffer is “always do action 1” and the new policy is “always do action 2”, resampling is impossible right?
>
> The replacement is possible with any policy or action. We hope Figure 4 in the updated submission clarifies that. Please let us know if it doesn't!
>
> > The discussions of actions that do not affect observations is too high level. A concrete example, even in a simple domain, could make the author’s point clear – can you provide such an example?
>
> Sure: With a constant 1-step delay, it takes 1 time-step to an action to be computed and sent, so the unaugmented observation captured at the end of this time-step is not influenced by the action.
> A better illustration is provided by Figure 3, which should make this high-level discussion straightforward. We referenced this figure in the updated submission to clarify the point.
>
> > Why is on-policy data even needed here?
>
> They are needed for multi-step value backups which are better than the 1-step backups done in SAC (check Lemma 1).
>
> > So can’t one compute the estimated states from the stored actions and provide that data to SAC? Why is resampling even needed?
>
> We are not sure what you mean by estimated states. Our paper is using MDPs and isn't involving any state estimation. Resampling is needed to create on-policy subtrajectories. Doing uncorrected multi-step updates on off-policy subtrajectories will produce biased value estimates. Our resampling is also better than importance sampling since it avoids variance explosion.
>
> > Firoiu et al.’s approach should appear in the empirical comparisons on constant-delayed MDPs
>
> We agree that a comparison with this work in the special case of constant delays would be interesting. Plus, we believe that it would be outperformed because their state-predictive approach adds a layer of uncertainty which DCAC doesn't suffer from. However, we were more interested in demonstrating that the methodology we propose works to improve off-policy algorithms such as SAC regardless of the delays being random or constant, than by comparing to the SOTA for constant delays, as constant delays barely exist in the real world.
>
> >Two references were missed
>
> Thank you for bringing these references to our attention. Both are now properly cited in the related work section, and compared to our approach. We also toned down the claim about being the first to deal with random delays.
>
> > The description of the prior work in the introduction as “proposing on-policy planning algorithms” does not make sense.
>
> We have removed "on-policy" in this sentence, as some approaches cited in the related work section are indeed off-policy.
>
> > The augmented space also causes significant exploration and generalization burdens.
>
> We agree and have added this. We want to stress though that this is not a negative of our approach. The augmented state space is simply how any Markov state in a delayed environment must look like.
>
> Previous work has often dealt with this approximately, which is also possible with DCAC by e.g. compressing the action-buffer in a hidden state. However, we didn't find an empirical need for doing this, even with delays over many time-steps (the WiFi total delays can go up to 9 time-steps).
>
> We hope that this addressed your most important concerns (in particular about the algorithm being difficult to understand). If so, we would really appreciate a reassessment. In any case we will be happy to hear your feedback.

---

> > ### Comment · AnonReviewer1 · 2020-11-17
> > **Additional question**
> >
> > Thank you to the authors for taking the time to update your paper and answer our questions.  While some things are clearer now I  (and I suspect other reviewers) have one more question about the resampling.  In  Figure 4, I now see that you are using the resampling to change the stored actions in the augmented states.  But it seems to me that could lead to invalid samples in he buffer.  Consider a number line environment with no noise and an agent that starts at 0.  Now suppose the agent's policy executes "left" twice leading to states -1 and -2.  Now suppose the agent decides it didn't like that and changes it's policy to "always right".  Your resampling operator appears to then replace the augmented actions with "right", so for instance we would have in the buffer an instance of [-2, <action>, right, right].  But it's not possible to reach -2 by taking two right actions from the origin, so this seems like an invalid sample.  Can the authors comment on how that specific case is avoided in their algorithm?

---

> > > ### Author Response · Authors · 2020-11-17
> > > **Great example! Adding delays will allow for resampling**
> > >
> > > Great! We think this example is very helpful. Generally, the action resampling can only be done in the presence of delays, so we will have to introduce delays to the example. For the example we will assume constant delays of one timestep which will require an action buffer $u_t$ containing the previously selection action. The full state $x$ would generally contain the position on the number line $s$, action delays $\alpha$, observation delays $\omega$ and the action buffer $u$ but since delays are constant, we will omit the delay components and say $x_t = (s_t, u_t)$.
> > >
> > > We still start at $x_0 = (0, \text{"stay"})$ where the initial action buffer contains the "stay" action. When agent executes "left" twice this will lead to $x_1=(0, \text{"left"})$ and then $x_2=(-1, \text{"left"})$. If the policy then changes to "always right" the constant delays of one will always allow us to produce on-policy trajectories of length one (+ start state) for the new policy. So if we start at the origin $x_0$ we can resample
> > >
> > > $x_1 = (0, \text{"left"})$ to $x^\*_1 = (0, \text{"right"})$
> > >
> > > and end up with the on-policy trajectory $x_0, x^\*_1$.
> > >
> > > ---
> > >
> > > Beyond that, it might also be useful to see what's going on for a constant delay of two timesteps (and action buffer of size two):
> > >
> > > We still start at $x_0 = (0, \text{"stay"}, \text{"stay"})$ where the initial action buffer contains two "stay" actions. When agent executes "left" twice this will lead to $x_1=(0, \text{"left"}, \text{"stay"})$ and then $x_2=(0, \text{"left"}, \text{"left"})$. If the policy then changes to "always right" the constant delays of two will always allow us to produce on-policy trajectories of length two  (+ start state)  for the new policy. So if we start at the origin $x_0$ we can resample
> > >
> > > $x_1 = (0, \text{"left"}, \text{"stay"})$ to $x^\*_1 = (0, \text{"right"}, \text{"stay"})$ and $x_2 = (0, \text{"left"}, \text{"left"})$ to $x^\*_2 = (0, \text{"right"}, \text{"right"})$
> > >
> > > and end up with the on-policy trajectory $x_0, x^\*_1, x^\*_2$.

---

> > > > ### Comment · AnonReviewer1 · 2020-11-19
> > > > **Re: example**
> > > >
> > > > Thanks for the details in the example.  The only thing I do not understand is that now the history buffer appears to contain invalid transitions.  x_2^* has been resampled to (0, right, right), which is fine, but the next observation in the history is going to be a "-1".  So won't the underlying model learn there is a possibility of transitioning from <0, right, right> to a state of <-1, ...>?  Note such a transition can't happen in the actual dynamics.  How are observations in the dataset updated if you change these actions?

---

> > > > > ### Author Response · Authors · 2020-11-19
> > > > > **Re: example**
> > > > >
> > > > > The model will not learn that there is a possibility to transition from <0, right, right> to <-1, ...>, and unaugmented observations in the dataset do not need to be updated.
> > > > >
> > > > > This is because <0, right, right> is necessarily the last resampled augmented state of the partially resampled trajectory (its action buffer has been completely resampled).
> > > > >
> > > > > The condition of Theorem 1 enforces this. Let us unroll the whole resampling procedure to illustrate the point:
> > > > >
> > > > > $K=2$
> > > > >
> > > > > $x_0 = (0, \alpha_0, \omega_0, \text{left}, \text{left})$ (initial state)
> > > > >
> > > > > $x_1 = (-1, \alpha_1, \omega_1, \text{left}, \text{left})$
> > > > >
> > > > > $x_2 = (-2, \alpha_2, \omega_2, \text{left}, \text{left})$
> > > > >
> > > > > $x_3 = (-3, \alpha_3, \omega_3, \text{left}, \text{left})$
> > > > >
> > > > > - $\alpha_1+\omega_1 \geq 1$ since we assume $\forall i,\alpha_i \geq 1$ ($\alpha_i$ is at least the inference duration). So the condition of Theorem 1 is satisfied for $x_1$, which we can therefore resample:
> > > > >
> > > > > $x^\*_1 = (-1, \alpha_1, \omega_1, \text{right}, \text{left})$
> > > > >
> > > > > - If $\alpha_2+\omega_2 \geq 2$, the condition of Theorem 1 is still true for $x_2$. In such case we can therefore also resample $x_2$:
> > > > >
> > > > > $x^\*_2 = (-2, \alpha_2, \omega_2, \text{right}, \text{right})$
> > > > >
> > > > > - Now, to resample $x_3$, we would need $\alpha_3 + \omega_3 \geq 3$ (otherwise the condition of Theorem 1 does not hold). However, $K=2$ (remember that the action buffer is of length $K$). Therefore $\alpha_3 + \omega_3$ must be $\leq 2$ and the condition of Theorem 1 cannot hold for $x_3$, which will never be resampled when $x_0$ is sampled in the replay memory as initial state of the trajectory fragment.
> > > > >
> > > > > So we never resample invalid transitions (and therefore we don't use them in the n-step backup performed by $\hat v(x_0, \tau^\*_n)$). Note that the resampled $x_1^\*$ and $x_2^\*$ are valid even though $\pi$ is "always right", not because of the policy but because of the action buffer of the initial state.
> > > > >
> > > > > PS: We have added a new figure to the paper (Figure 5) based on this example. This is likely to answer your question, and, along with Figure 4, we hope this also addresses your initial concern about needed illustrations in order to make it easy to understand the algorithm. We would appreciate your feedback regarding this change.

---

> > > > > > ### Comment · AnonReviewer1 · 2020-11-24
> > > > > > **Re: example**
> > > > > >
> > > > > > Thank you, that example illustration helped.  I assume that under non-constant conditions re-sampling can still happen after the first few actions?  It would be good to update theorem 1 to make that clear, and specifically define "t" in theorem 1 since it is a free variable now.

---

> > > > > > > ### Author Response · Authors · 2020-11-24
> > > > > > > **t is now defined explicitly in Theorem 1**
> > > > > > >
> > > > > > > Under constant conditions, the resampling always happens over the full length of the action-buffer (because under constant conditions the total delay is always $K$).
> > > > > > >
> > > > > > > Under non-constant conditions, the length of the resampling depends on the trajectory fragment sampled from the replay memory.
> > > > > > > The average resampling length depends on the delay distributions, but even with uniform distributions (c.f. Appendix) it is often fairly long.
> > > > > > > This is because the condition of Theorem 1 starts very loose (for the first action of the buffer it is $\alpha_1 + \omega_1 \geq 1$, i.e. 'True') and it tightens as we advance in the trajectory fragment (for the last action of the buffer it is $\alpha_K + \omega_K \geq K$, i.e $\alpha_K + \omega_K = K$).
> > > > > > >
> > > > > > > In Theorem 1, $t$ is not really a free variable, it is an index that goes from $1$ to $n$ in a given trajectory fragment $\tau^\*_n$ (the condition must hold for each $\alpha^\*_t, \omega^\*_t$ in $\tau^\*_n$ for the theorem to hold for $\tau^\*_n$). We have updated the submission to define $t$ explicitly in Theorem 1.

---

### Official Review · AnonReviewer4 · 2020-10-27
**Interesting setting with straightforward solution**

**Rating:** 6
**Confidence:** 3

**Review:**

Post rebuttal: The updates clearly explain the resampling procedure of this paper, and strengthen the theoretical part of this paper. As a result, I'd like to change my rating to 6 and recommend an acceptance.

Additional thoughts emerged from the discussion with authors (which are irrelevant to the rating): I agree that experiments in the paper demonstrate the modified SAC algorithm has a significant improvement compared to baseline algorithms. And I believe that the paper could benefit from including some theoretical justifications to the loss function and data collection scheme (though I completely understand the difficulty of theoretically justify deep RL algorithm). For example:
- Assuming all the loss functions can be optimized to optimal, will the policy converge to optimal or near-optimal solutions?
- Assuming the value net can be optimized to optimal, how the resampling process change the gradient of policy net? In which case would the on-policy sample with truncated trajectory (i.e., the value function computed by Eq. (8) where the length of the trajectory is $n$) out-perform off-policy sample with full trajectory (i.e. SAC without resampling)? If I understand correctly, without resampling the error of the value net suffers from the amplification caused by distribution mismatch (which is potentially exponential?). And with resampling, would the error of value net come from the truncation?

Additional minor issues:
- Definition 5: There is no base (i.e., $n=0$) in the recursive definition of $\hat{v}_n^{soft}$.

---------
The paper considers reinforcement learning with delays, motivated by real-world control problems. Novelty of the setting is that the delay is random and changing. Algorithm proposed by the paper uses importance sampling to create on-policy samples of augmented observations, and is empirically shown to out perform base line SAC algorithm.

While the idea of the paper is clean, I found it a bit hard to follow the complicated notations and definitions without intuition explanation. I appreciate the effort of making the paper mathematically rigorous, but I believe that the paper could be more easy-to-follow and have a larger impact to the community if there were more explanations before/after each definition, especially when the math behind this paper is not super complicated.

Some additional questions:
1. What is the observation if the delay decrease by more than one? If I understand correctly, does Eq. (1) imply that the agent can only observe the last state? In other words, suppose there are some delay of the network such that there are no observations for 5 time steps, and after that all the network packages arrive at the same time. Will the agent discard the information of time steps 1, 2, 3 and 4?
2. In order to have a Markovian transition, definition 1 requires $K$ being the maximum possible total delay. However, Theorem 1 assumes that total delays are longer than the trajectory. Does it imply that $w_i+\alpha_i$ is a constant? Otherwise at least one of the assumptions can not be true.
3. I couldn't follow the proof of Theorem 1 (and Lemma 6). In Definition 3, Eq. (2), what is $u_0^*$? For the induction in Lemma 6, what is the induction base? If I understand correctly (please correct me if I'm wrong), the operator $\sigma_n^\pi(\tau_n^\star\mid x_0;\tau_n)$ is similar to probability ratio in standard importance sampling method, and can only assign non-zero value to trajectories $\tau_n^\star$ such that $s_i^\star=s_i,\forall i$. If this is the case, I'm not convinced that Eq. (3) can hold. For example, there might be a sequence $\tau_n^\star$ such that $p_n^\mu(\tau_n\mid x_0)=0$ for every $\tau_n$ such that $s_i=s_i^\star$. And policy $\pi$ can reach such sequence (i.e., $\pi_n^\pi(\tau_n^\star\mid x_0)>0$). Will it violate Eq. (3)?

In summary, I believe that the RL with delay setting is important and interesting. However, due to over-complicated notations and theorem statement, I'm not able to verify the soundness/correctness of the method. I can not recommend acceptance at this point, and I'm willing to discuss and change my score if my main concerns are answered.

---

> ### Author Response · Authors · 2020-11-14
> **Thank you for you in-depth review and your diligence to go through our proofs**
>
> We thank you for you in-depth review and your remarkable diligence to go through our proofs. We appreciate your good faith in this discussion process and are willing to do our best in order to address your concerns.
>
> >  I found it a bit hard to follow the complicated notations and definitions without intuition explanation
>
> As the reviews show we overestimated how much the RDMDP definition would speak for itself. Ironically the exact definition of the RDMPD is not even needed to understand our main algorithmic contribution and theorems, only the proofs. An intuitive understanding as we tried to convey via Figure 3 is perfectly adequate.
>
> > What is the observation if the delay decrease by more than one? If I understand correctly, does Eq. (1) imply that the agent can only observe the last state? In other words, suppose there are some delay of the network such that there are no observations for 5 time steps, and after that all the network packages arrive at the same time. Will the agent discard the information of time steps 1, 2, 3 and 4?
>
> You are correct, yes. The reason why the agent discards this information is that it is already compressed in the most recently produced observation available (Markov observations).
>
> In the situation you describe, the agent will repeat 5 times the observation available at time-step 0 (which is why the delay cannot increase by more than 1). Then, it will select the most recently produced observation in the received packets at time-step 5, and discard others since they only contain outdated information.
>
> > In order to have a Markovian transition, definition 1 requires $K$ being the maximum possible total delay. However, Theorem 1 assumes that total delays are longer than the trajectory. Does it imply that $\omega_i + \alpha_i$ is a constant? Otherwise at least one of the assumptions can not be true.
>
> There is an important confusion here. Theorem 1 does not assume the total delays to be longer than the trajectory. It assumes the total delays to be longer than the past part of the considered trajectory fragment. In other and simpler words, it assumes that none of the actions computed from any state of this trajectory fragment had an effect on any subsequent (unaugmented) observations of the so-called trajectory fragment. We have rephrased Theorem 1 in order to prevent this confusion.
>
> So, of course it does not imply that $\omega_i + \alpha_i$ is constant: this would have made our algorithm quite pointless for random delays.
>
> > In Definition 3, Eq. (2), what is $u^\*_0$?
>
> $u^\*_0$ is the action buffer contained in $x^\*_0$, given recursively as an input to $\sigma$. Since all reviewers found this definition difficult to understand, we have added a visual illustration of the resampling procedure (Figure 4 in the updated submission). This should make it much easier to understand. We would appreciate your feedback on this.
>
>
> > If I understand correctly (please correct me if I'm wrong), the operator $\sigma^\pi_n(\tau^\*_n | x_0, \tau_n)$ is similar to probability ratio in standard importance sampling method, and can only assign non-zero value to trajectories $\tau^\*_n$ such that $s^\*_i = s_i, \forall i$.
>
> It should now be clear from Figure 4 in the updated submission that the resampling operator $\sigma^\pi_n(\tau^\*_n | x_0, \tau_n)$ is nothing like the probability ratio in importance sampling. It can clearly assign zero probability to such trajectories, even if $(s^\*_i, \omega^\*_i, \alpha^\*_i, r^\*_i) = (s_i, \omega_i, \alpha_i, r_i), \forall i$, because all it does is resample actions $a_i$ in $\pi$. If one $\pi(a_0|x^\*_0)=0$ (note that $a_0$ recursively points to all resampled actions in equation (2)), then the expectation  is $0$ and $\sigma^\pi_n(\tau^\*_n | x_0, \tau_n) = 0$.
>
> >If this is the case, I'm not convinced that Eq. (3) can hold. For example, there might be a sequence $\tau^\*_n$ such that $p^\mu_n(\tau_n | x_0) = 0$ for every $\tau_n$ such that $s_i=s^\*_i$.And policy $\pi$ can reach such sequence (i.e., $p^\pi_n(\tau^\*_n | x_0) > 0$). Will it violate Eq. (3)?
>
> We don't think this is a possibility. If $p^\mu_n(\tau_n | x_0) = 0$ for every $\tau_n$ such that $s_i=s^\*_i$ then the sequence of unaugmented observations $s_i=s^\*_i$ is impossible in the environment for any policy. This is true because $\mu$ has no effect on any $s_i$ because the actions sampled from $\mu$ are delayed and will only start to have an effect starting with observation $s_{n+1}$. Since $\mu$ has no effect in the statement you can exchange $\pi$ for $\mu$. Therefore $p^\pi_n(\tau^\*_n | x_0)$ must be zero.
>
>
> > For the induction in Lemma 6, what is the induction base?
>
> Indeed, we forgot to include the induction base here! We updated the paper to include it, as you can see it is fairly trivial.

---

> > ### Comment · AnonReviewer4 · 2020-11-18
> > **Thanks for the clarification!**
> >
> > Thank you for the detailed clarification (especially Figure 4). The resampling process is now clear to me.
> >
> > I have a follow up question on the relation between $K$ and $w,\alpha.$ Could the author clarify the relation between the following three terms: the maximum total delay $K$, the length of the sampled trajectory $n$ (as defined in the last paragraph of page 5), and the number of observations (denoted by $S$) in one state (say $x_0$ in Eq. (9))? In particular, is it true that
> > 1. $K\ge w_{n-1}^\star+\alpha_{n-1}^\star\ge n-1,$ (the assumption of Theorem 1)
> > 2. $S\le n,$ and
> > 3. $S\ge w_{n-1}^\star+\alpha_{n-1}^\star$ (in order to make the transition Markovian)?
> >
> > Additional follow up questions:
> > - Whether the number of observations in one state a constant?
> > - What is the architecture for critic $v_\theta(x_0)$, is it a RNN-like structure because $x_0$ has more than one observation?

---

> > > ### Author Response · Authors · 2020-11-19
> > > **Thank you for your feedback**
> > >
> > > Thank you for your feedback regarding the changes we made in the paper.
> > >
> > > Regarding your follow-up question, we are not sure what you mean by "the number of observations (denoted by $S$) in one state (say $x_0$ in eq (9))".
> > >
> > > In the paper, we use $S$ to denote the unaugmented state-space in Definition 1 (we have updated the paragraph after Definition 1 to make this notation explicit).
> > >
> > > According to Definition 1, an augmented state $x_i$ is formed by an unaugmented state $s_i \in S$, an action delay $\alpha_i \in \mathbb{N}$, an observation delay $\omega_i \in \mathbb{N}$ and an action buffer $u_i \in A^K$ containing the $K$ last computed actions: $x_i=(s_i, \alpha_i, \omega_i, u_i)$. There is no varying number of observations in this definition, so we are not sure what you mean when redefining $S$.
> > >
> > > In equation (9) (Definition 6), $x_0$ is the first augmented state in a trajectory $(x_0, \tau_n)=(x_0, x_1, x_2, ..., x_n)$ sampled from the replay memory (in fact, $\tau_n$ also contains the rewards but let us omit them here to keep the notation light).
> > >
> > > > $K \geq \omega_{n-1}^\* + \alpha_{n-1}^\* \geq n-1$
> > >
> > > You are correct, yes, this is true by definition of $K$ and $n$:
> > >
> > > As defined in Definition 1, $K$ is the maximum possible total delay (the total delay of $x_i$ is $\alpha_i + \omega_i$).
> > >
> > > Delays are not modified by $\sigma$, so $\alpha^\*_i + \omega^\*_i = \alpha_i + \omega_i \leq K$.
> > >
> > > As defined in the last paragraph of page 5, given an augmented state $x_0$ sampled from the replay memory, $n$ is the largest number such that the subsequent augmented states $x_1, x_2, ..., x_n = \tau_n$ have only action delays $\alpha_i$ and observation delays $\omega_i$ that satisfy the condition of Theorem 1.
> > >
> > > So for every $i \leq n$ we have $K \geq \omega^\*_i + \alpha^\*_i \geq i$.
> > >
> > > > $S \leq n$
> > >
> > > With the notation of the paper, this equation does not make sense because $S$ is a state-space and $n$ is a scalar. As explained above, we are not sure what you mean when redefining $S$.
> > >
> > > > $S \geq \omega_{n-1}^\* + \alpha_{n-1}^\*$
> > >
> > > Same as above.
> > >
> > > > Whether the number of observations in one state is a constant?
> > >
> > > In one augmented state, there is 1 unaugmented state, 1 action delay, 1 observation delay, and an action buffer of $K$ actions (c.f. Definition 1).
> > >
> > > > What is the architecture for critic $v_\theta(x_0)$, is it a RNN-like structure because $x_0$ has more than one observation?
> > >
> > > This can be any architecture, but in our experiments we concatenate all the parts of $x_0$, i.e. $s_0, \alpha_0, \omega_0$ and $u_0$, into a single vector and feed this vector to a multilayer perceptron.
> > >
> > > The precise architecture of the model used in our experiments is described in Appendix B.2.
> > >
> > > PS: We have updated the paper with a new figure that unrolls the whole algorithm described in Section 3 on a simple example (Figure 5). This figure is likely to answer your question. Please let us know whether it does.

---

> > > > ### Comment · AnonReviewer4 · 2020-11-20
> > > > **Thanks for answering my questions!**
> > > >
> > > > Thanks for the detailed answers! My main concerns are all addressed, and I updated my review accordingly.

---

> ### Author Response · Authors · 2020-11-24
> **Answer to additional comments**
>
> Thank you for updating your review! We want to briefly address your additional comments.
>
> > Assuming all the loss functions can be optimized to optimal, will the policy converge to optimal or near-optimal solutions?
>
> It's not even really clear what "optimal loss" means in this case because the loss functions change as the models change. But if we say that we are perfectly approximating the value function then this should be equivalent to policy gradient and the policy should be guaranteed to converge to a local optimum.
>
> > Assuming the value net can be optimized to optimal, how the resampling process change the gradient of policy net?
>
> If we assume that our value net is the true value function then the our policy loss and SAC's policy loss are the same. The advantage comes only from reducing bias in the value approximation.
>
>
> Lastly, Definition 5 has been fixed to include the base-case.

---

### Official Review · AnonReviewer3 · 2020-10-29
**The paper improved Soft-Actor-Critic algorithm in more realistic environments with random action/state delays.**

**Rating:** 8
**Confidence:** 4

**Review:**

This paper studies the effects of random action + observations in reinforcement learning, and proposed to use partial trajectory resampling to improve off-policy algorithms such as SAC. The proposed method performs strongly in multiple environments augmented with random delays, and beats baseline SAC and RSAC in both sample efficiency and final policy performance.


Pros:

(1) The methodology is sound, and the performance gaps between the proposed method and baselines are significant, especially when the tasks become harder as delay increases.

(2) The experiments are comprehensive and demonstrate the potential of this algorithm.

Cons:

(1) The paper introduced a critical concept: partial trajectory resampling in Sect 3.1. I find that this section is generally not easy to read given the packed symbols. In addition to the equations, it would be better to add a figure illustrating the recursive sub-sampling process.

(2) The paper only augmented SAC. However, in theory this paper should apply to other off policy learning algorithms such as TD3. It would be more comprehensive to try such studies.

(3) Clearly the baseline SAC suffers in environments with large latencies and the learning is slow. In these scenarios I am interested in seeing an on-policy baseline such as PPO/TRPO, which generally seems to be more robust to state/action delays.


Conclusion:
Please address my concerns raised in the "cons" section.

---

> ### Author Response · Authors · 2020-11-14
> **Thanks for taking the time to understand our paper and for your pertinent feedback**
>
> We thank you for taking the time to understand our paper and for your pertinent feedback. We answer your concerns below.
>
> (1): You are right to note that the partial trajectory resampling operator is critical to the understanding of the paper, and that providing only its fairly heavy mathematical formulation without a visual illustration was not making it easy. Therefore, we added the requested figure to the paper (see general comment). We also reformulated some sentences in section 3.1 in hope to make this section easier to follow. We would appreciate your feedback regarding these changes.
>
> (2): Rather than just DCAC, we indeed provide a methodology that can be adapted to improve many off-policy algorithms such as TD3 in delayed settings, and we agree that it would be interesting to test this approach on more algorithms than just SAC. The reason why we chose to improve SAC in particular is that it is currently the best-performing SOTA algorithm in a fair range of robotic applications, where the maximum entropy framework has interesting properties e.g. in terms of robustness. Indeed, we designed this approach with real-world applications in mind as future work. As you can imagine, setting up sound experiments in Delayed MuJoCo with e.g. TD3 instead of SAC will cost time and computational resources, but we would like to see such other applications in the future.
>
> (3): Our partial resampling approach essentially transforms the biggest possible slices of off-policy trajectories into on-policy sub-trajectories (which is only possible in delayed settings). This allows us to keep the sample-efficiency of off-policy algorithms such as SAC while compensating for the multi-step credit assignment difficulty introduced by the delays (as an on-policy algorithm such as PPO would do). In this sense we expect it to outperform PPO in robotic settings where data collection is costly, since SAC already outperforms PPO in such settings even without the delay correction that we introduce in DCAC.

---

### Author Response · Authors · 2020-11-11
**Visual explanation of Definition 3, and code release**

We thank all the reviewers for their detailed and constructive feedback.

We are sorry to find that R2 could not follow the mathematical development of our paper. Since all other reviewers have understood our main theoretical contributions, we sincerely hope that the discussion phase will help clear this misunderstanding. We realize that after working a lot with our mathematical notations and recursive equations they are very natural for us, but may not be as straightforward to understand from an external point of view.

Therefore, it seems that the most important thing to do first is to give a visual explanation of the resampling operator of Definition 3, as requested directly or indirectly by R1, R2, R3 and R4 altogether. Indeed, understanding how this operator works is crucial in order to understand the point that Theorem 1 is making, i.e. our main theoretical contribution.

The paper has been updated with a first version of the requested figure (Figure 4), also provided [here](https://i.imgur.com/8TrVtaV.png) for convenience.

As described in this figure, the principle of the partial resampling operator is actually very simple: it recursively resamples actions in the action buffers, replacing them by on-policy actions. Its recursive definition as a conditional distribution in Definition 3 might appear a bit scary (as for RDMDPs), but we needed these definitions to formally prove the theorem. In Definition 3, $\delta(x^*|x)$ is the delta Dirac distribution which, when marginalized, simply says that $x^*$ is $x$ with probability 1.

Although we cannot yet open our github project to respect the anonymity of the reviewing process, we link here an anonymous copy of our current implementation of DCAC, which may help clarify possible confusions and clear reproducibility concerns (we will open-source the entire project along the final paper).
The resampling operator of Definition 3 is implemented in the 'for' loop of line 89:

https://pastebin.com/yTjwYs76
password: dcac

In hope that this answered the general request for a visual description of Definition 3, we will now work on addressing individual concerns during the next few days.

---

### Author Response · Authors · 2020-11-19
**New figure in the updated submission: main algorithm unrolled on a simple example**

We thank the reviewers for their involvement in the discussion process and for their interesting feedback.

We are happy to find that Figure 4, which we added at the beginning of the discussion phase to visually explain Definition 3, successfully clarified the partial resampling operator for the reviewers.

Therefore, we have now updated the submission with the first version of another complementary figure (Figure 5), which visually unrolls the whole algorithm described in Section 3 on a simple example of 1D-world with random delays, inspired from the discussion with R1. This figure is explained in the last paragraph of page 5.

We think that this example will make the main algorithm and our notation easier to understand for the new reader, therefore giving a bigger impact to the paper.

We would appreciate your feedback regarding this new addition to the paper.

---

### Author Response · Authors · 2020-11-25
**Authors thanks and conclusion**

As the discussion phase with us ends, we want to thank the reviewers again for their involvement and for their interesting feedback.

In particular, the discussion yielded changes in different parts of the text to make the paper easier to follow, some relevant additions to the related work section, and the addition of Figures 4 and 5 (which we believe will be helpful for the new reader, as the reviewers found these informative).

The discussion with R1 and R4 was particularly constructive. We thank them for their involvement and humbly believe that their concerns have now all been addressed.

R3 didn't have difficulties understanding the paper and we thank them for their straightforward review. We hope our response correctly answered their few concerns.

---

### Decision · Program_Chairs · 2021-01-07
**Final Decision**

**Decision:**

Accept (Poster)

**Comment:**

This paper considers the RL problems where actions and observations may be delayed randomly. The proposed solution is based on generating on-policy sub-trajectories from off-policy samples. The benefits of this approach over standard RL algorithms is clearly demonstrated on MuJoCO problems. The paper also provides theoretical guarantees.
This paper is well-written overall and technically strong. The majority of the reviewers find that this paper would constitute a valuable contribution to the ICLR program.